# CoNNect: A Swiss-Army-Knife Regularizer for Pruning of Neural Networks

## Abstract

Pruning encompasses a range of techniques aimed at increasing the sparsity of neural networks (NNs). These techniques can generally be framed as minimizing a loss function subject to an $L_0$-norm constraint. In this paper, we introduce CoNNect, a novel differentiable regularizer for sparse NN training that quantifies connectivity in weighted graphs. Our theoretical and numerical analyses show that CoNNect integrates seamlessly with many established pruning strategies and is applicable to both unstructured and structured pruning. By including CoNNect as a regularizer during training, we ensure neural networks maintain connectivity between input and output layers, addressing limitations of $L_1$-regularization, a common surrogate for $L_0$-norm regularization. We prove that CoNNect effectively approximates $L_0$-regularization, guaranteeing maximally connected network structures as stable stationary points and avoiding issues like layer collapse. Through numerical experiments, we demonstrate that classical pruning strategies benefit from CoNNect regularization compared to $L_1$- and $L_2$-norm regularization. Additionally, we show that integrating CoNNect into LLM-pruner, a one-shot pruning method for large language models, yields improved results.

## 1 Introduction

This paper aims to investigate the creation and enhancement of a sparse neural network (NN). Sparse NNs, known for their drastic reduction in the number of active connections or parameters, have attracted significant interest in recent years due to their ability to boost computational efficiency and minimize memory consumption while preserving or even improving model performance (LeCun et al., 1989; Hassibi et al., 1993; Frankle & Carbin, 2018).

To achieve sparsity in neural networks, various techniques have been proposed and applied in different domains. For example, weight pruning (Hagiwara, 1993), neuron pruning (Huang & Wang, 2017), and structured pruning (e.g., see (Yuan & Lin, 2006; Anwar et al., 2017)) are common methods used to reduce model size. Pruning refers to the process of systematically eliminating parameters that contribute little to network performance, effectively simplifying the model. By carefully identifying and removing these less critical components, the resulting sparse network retains its ability to make accurate predictions while benefiting from increased efficiency.

We believe that pruning should obey the following two axioms (where we identify a NN with a directed, weighted graph):

**Axiom 1** (Delete as Many Weights as Possible). *For the point of memory and energy consumption, the graph should be "small": in pruning we want to drastically reduce the number of edges and maybe even nodes while only mildly affecting accuracy.*

**Axiom 2** (Preserve Neural Network Connectivity). *The pruning process must prevent disruptions in the connectivity of the neural network and preserve the flow of information from input to output.*

The extensive research on pruning neural networks, as more elaborately outlined in the literature overview in Section 2 and particularly in review works such as Hoefler et al. (2021); He & Xiao (2023), predominantly aligns with the first axiom. However, few methods address Axiom 2, as the impact of weight removal on overall network connectivity is rarely considered. This negligence can result in a pruning that produces highly disconnected networks, or in the most extreme case so-called

*layer collapse*, see Figure 1, where the NN becomes completely dysfunctional. A notable exception is SynFlow pruning (Tanaka et al., 2020), which we will explore in more detail in Section 3.3.2.

In this paper, we propose a new regularizer, called CoNNect, that can be used to satisfying both axioms simultaneously and (i) is differentiable (except in the point zero) and allows for gradient descent optimization, (ii) effectively approximates $L_0$-regularization and guarantees maximally connected network structures as stable stationary points, avoiding issues such as layer collapse. CoNNect is based on the Katz centrality measure (Katz, 1953), which evaluates the connectivity of weighted graphs by utilizing the connectivity measurement employed by Katz centrality for networks with normalized weights. Normalization results in weights being restricted to $[0, 1]$, so that the contribution of path from input to output layer to the over-

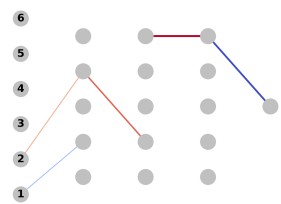

Figure 1: Layer collapse in a pruned NN.

all connectivity of the network goes exponentially quick to zero unless the weights along the paths are (close to) 1. Hence, when maximizing the connectivity for the normalized weights, we find a weight association that prefers few "direct paths" over many "parallel paths", while focusing on connectivity of the input with the output layer. As is clear form the above, including the CoNNect regularizer in training of an NN, leads in a natural way to a sparse network representation, and hence satisfies Axiom 1 and Axiom 2 simultaneously.

CoNNect is a versatile regularizer that can be integrated in many established pruning strategies. Moreover, it is suitable for both unstructured and structured pruning. We demonstrate its efficacy through a series of numerical examples. First, we show in an unstructured pruning example that pruning strategies like magnitude-pruning (LeCun et al., 1989; Hassibi et al., 1993) and SynFlow (Tanaka et al., 2020) can benefit from CoNNect regularization during training. Here, it outperforms $L_1$ and $L_2$ regularization in terms of both accuracy and stability. For structured pruning, we apply CoNNect at the channel level of VGG-11 and a Graph Neural Network (GNN), achieving improved performance compared to $L_1$ regularization. Moreover, we integrate CoNNect into LLM-pruner (Ma et al., 2023), a one-shot pruning method for Large Language Models (LLMs), and show improved results. We believe this versatility positions CoNNect as a promising framework for future exploration and development in neural network pruning strategies.

## 2 RELATED WORK

The concept of pruning NNs dates back to the early 1990s. The seminal work by LeCun et al. (1989) on Optimal Brain Damage introduced the idea of pruning by removing weights that contribute least to performance, thus simplifying the network. Hassibi et al. (1993) extended this concept with Optimal Brain Surgeon, which provided a more sophisticated method for determining which weights to prune based on their impact on the error function. These early methods laid the foundation for modern pruning techniques, focusing on reducing network complexity while maintaining accuracy.

**Unstructured vs. Structured Pruning.** Pruning methods can be broadly categorized into unstructured and structured pruning. Unstructured pruning involves selectively removing individual weights from the network. Unstructured pruning can lead to highly sparse networks, but often results in irregular memory access patterns, which can be difficult to optimize in hardware implementations. Pruning neural network weights based on absolute values is a classic example of unstructured pruning (LeCun et al., 1989; Hassibi et al., 1993; Hagiwara, 1993; Han et al., 2015). This method is effective in reducing the number of active parameters, but may not always lead to practical improvements in computational efficiency. In contrast, structured pruning removes entire groups of parameters, such as neurons, filters, or even layers. This approach results in a network structure that is more amenable to efficient hardware implementations. Techniques like Group Lasso (Yuan & Lin, 2006; Hoefler et al., 2021) and other structured sparsity learning (Wen et al., 2016; Zhuang et al., 2020) fall into this category; see He & Xiao (2023) for a review. Structured pruning tends to preserve the regular structure of the network, which can lead to greater practical efficiency improvements, though it may require more careful consideration to avoid significant loss of accuracy.

**Regularization-Based Pruning (Soft Pruning).** Regularization methods play a crucial role in promoting sparsity during the training process by extending the loss function with a penalty function that discourages overly complex models. While sparsity is encouraged, regularization does not

explicitly set the weights to zero but instead reduces their magnitude, allowing them to remain non-zero and potentially become active again if needed. This leads to what is termed soft pruning, where sparsity is encouraged but not strictly enforced through hard weight removal during training. It is only that after training unimportant weights are pruned, typically via pruning the smallest weights in magnitude Hagiwara (1993); Gale et al. (2019). One of the simplest and most widely used methods, L1-regularization (Tibshirani, 1996; He et al., 2017; Yang et al., 2019; De & Doostan, 2022; Ziyin & Wang, 2023), penalizes the sum of the absolute values of the weights, encouraging many weights to become zero. Moreover, L1-regularization fails to incorporate considerations from Axiom II, which emphasizes the preservation of neural network connectivity and functionality. This lack of consideration for connectivity can lead to a network that, while sparse, may suffer from disrupted information flow, ultimately impairing its performance. Similarly, L2-regularization, another common regularization technique, penalizes the sum of the squares of the weights (e.g., see Hinton (2012); Phaisangittisagul (2016); Loshchilov et al. (2017)). While L2-regularization is effective at discouraging large weights, it does not push small weights towards zero, thus failing to induce sparsity in the network. As a result, L2-regularization typically produces networks with small but non-zero weights, which do not benefit from the same computational efficiency gains that a sparse network would offer. Moreover, like L1-regularization, L2-regularization does not address the need to maintain critical connections as highlighted by Axiom II, making it less suitable for tasks where maintaining network connectivity is essential.

**Stage-Based Pruning (Hard Pruning).** Stage-based pruning strategies are utilized as separate, discrete actions during various stages of model training. These techniques can be implemented before training (Lee et al., 2018; Tanaka et al., 2020; Wang et al., 2020), during training (Frankle & Carbin, 2018), or after training (Hagiwara, 1993; Thimm & Fiesler, 1995; Gale et al., 2019; Ma et al., 2023). Stage-based pruning generally does not fundamentally alter the objective function or the descent direction like regularization does, but instead acts on the model's structure or parameters at specific moments. These kind of pruning methods can be considered hard pruning approaches, as parameters are explicitly removed. Many different criteria for pruning have been introduced, such as magnitude-based pruning (Hagiwara, 1993; Gale et al., 2019), which involves removing weights with the lowest absolute values and is based on the idea that these weights have the least impact on the overall performance of the model. More complex criteria have been constructed to determine the impact of weight removal, such as first-order (e.g., see (Zhou & Si, 1999; Molchanov et al., 2016; Sanh et al., 2020)) and second-order expansions (LeCun et al., 1989; Hassibi et al., 1993; Ma et al., 2023) of the training objective. Specifically, SynFlow (Tanaka et al., 2020) is a method that adheres closely to the principles of Axiom II, focusing on retaining the network's connectivity and functionality during pruning. Unlike magnitude-based techniques, SynFlow utilizes a first-order expansion of signal flow to pinpoint and remove weights with minimal impact on the network's overall information flow. This approach ensures that while the network is being pruned, its structural integrity is preserved and the critical pathways in terms of connectivity remain intact.

We conclude the above discussion by noting that the CoNNect regularizer, to be introduced in the next section, can be integrated in any of the above stage-based pruning approaches.

## 3 METHODOLOGY

### 3.1 PRELIMINARIES

We define a graph $\mathcal{G} = (V, E)$, where $V$ denotes the set of vertices (or nodes) and $E$ represents the set of directed links that connect these vertices. A weighted graph has weights $W_{i,j} \geq 0$ for links $(i, j) \in E$, where we let $W_{i,j} = 0$, for $(i, j) \notin E$. Neural networks can be described using graph theory by representing them as directed, weighted graphs. In this setting, the vertices $V = V_1 \cup \ldots \cup V_K$ in the graph correspond to the neurons in the network which are organized into distinct subsets corresponding to the different layers $V_k$, for $k = 1, \ldots, K$. Here, the input nodes $V_1$ represent the neurons in the input layer, the hidden nodes $V_k$, for $k = 2, \ldots, K-1$, represent the neurons in the hidden layers, and the output nodes $V_K$ represent the neurons in the output layer.

For simplicity of the ensuing analysis, we assume a simple feedforward neural network without skip connections, so that each pair of subsequent layers $V_k$ and $V_{k+1}$ is connected via edges in the set $E_k$, for $k = 1, \ldots, K-1$. While this formulation excludes possible skip connections, we discuss

later that architectures with skip connections, i.e., residual neural networks, can also be regularized using CoNNect.

Throughout the paper, we describe a neural network $\mathcal{G}$ using the tuple $(W, b)$, where $W \in \mathbb{R}^{|V| \times |V|}$ is the weighted adjacency matrix of the weights, such that $W_{i,j}$ connects node $i \in V_k$ with node $j \in V_{k+1}$, and $b = (b_1, \ldots, b_{|V|})$ is the bias vector. Moreover, we denote the activation of the $k + 1$th layer by the tensor

$$X^{(k+1)} = \sigma\big(W^{(k)} X^{(k)} + b^{(k+1)}\big),$$

where $\sigma$ is the activation function, $W^{(k)}$ is the submatrix containing the weights between nodes in $V_k$, and $V_{k+1}$, and $b^{(k+1)}$ the biases for the nodes in $V_{k+1}$. Finally, we denote $f(X^{(1)}; W, b)$ as a forward pass through the network.

## 3.2 PROBLEM FORMULATION

Let $\{(x_i, y_i)\}_{i=1}^N$ denote the training set, where $x_i = X_i^{(1)}$ represents the input data and $y_i$ represents the corresponding label for each of the $N$ samples. Fitting the parameters of a neural network $\mathcal{G}$ involves optimizing the network's weights to minimize a loss function $\mathcal{L}(\hat{y}, y)$, where $\hat{y} = f(x; W, b)$ is the predicted output given an input $x$.

In this paper, our objective is to train a sparse neural network, which can be achieved by inducing sparsity in the network's parameters. A commonly employed approach to sparsification is regularization. Regularization involves augmenting the loss function with an additional term that penalizes non-zero elements in the network parameters. Specifically, the optimization problem can be formulated as:

$$\min_{W,b} \quad \mathcal{L}(\hat{y}, y) + \lambda R(W), \tag{1}$$

where $R(W) = \|W\|_{0,1}$. However, this $L_0$-norm is non-convex and leads to a combinatorial optimization problem, which is generally NP-hard and computationally intractable for large-scale problems. A more practical alternative is $L_1$-regularization, as in Lasso regression, where $R(W) = \|W\|_{1,1}$. $L_1$-regularization induces sparsity by shrinking weights to zero, approximating the $L_0$-norm while remaining convex and suitable for gradient-based optimization. However, $L_1$-regularization primarily satisfies Axiom 1 by reducing connections but fails to address Axiom 2, which focuses on preserving network connectivity and ensuring efficient signal flow. This limitation can result in a disconnected or underperforming network when key pathways are not maintained.

## 3.3 CONNECT

To overcome the aforementioned issues, we propose CoNNect, a regularizer that considers both individual weights and the network's overall connectivity, ensuring that the structure contributes to optimal performance. We first introduce CoNNect for unstructured regularization, along with some suitable hard pruning strategies. Then, we demonstrate how CoNNect can be seamlessly extended to structured regularization.

### 3.3.1 WEIGHT-LEVEL REGULARIZATION

Katz centrality is a measure used in network analysis to determine the relative connectivity of a node in a network by considering both the number and the quality of connections (Katz, 1953). Inspired by the connectivity measurement in Katz centrality, let us consider the following connectivity matrix for a network:

$$\varphi(W) = \sum_{k=1}^K (\theta(W))^k,$$

where $(\varphi(W))_{i,j}$ indicates the connectivity from node $i$ to node $j$, and $\theta(W)$ is a simple normalization of the network weights between two subsequent layers, e.g., for $i \in V_k$ and $j \in V_{k+1}$,

$$(\theta(W))_{i,j} = \frac{|W_{i,j}|}{\sum_{k \in V_k} \sum_{l \in V_{k+1}} |W_{k,l}|}. \tag{2}$$

In the context of a neural network, we can denote the connectivity by taking the sum of connectivity values between the input and output layer:

$$\varphi^{tot}(W) = \sum_{i \in V_1} \sum_{j \in V_K} (\varphi(W))_{i,j} \,.$$

Finally, we argue for the preservation of connectivity (as per Axiom 2), so we aim to maximize the network's overall connectivity. Consequently, we choose the regularizer as:

$$R(W) = -\varphi^{tot}(W), \tag{3}$$

which we will refer to as the CoNNect regularizer. A possible extension of CoNNect would be to include the biases and activation functions but, we leave this for future work.

CoNNect is effectively the (negative of the) sum of all (multiplicative) reparameterized weighted paths between nodes in the input layer $V_1$ and the output layer $V_K$. It follows that $-\varphi^{tot}(W) = 0$ if and only if there is no path with positive weight between the input and output layer. Moreover, $-\varphi^{tot}(W)$ can be efficiently computed using a single forward pass $f(\bar{1}, W, \bar{0})$, where $\bar{1}$ is a vector of ones as input, $\bar{0}$ is a vector of zeroes for the biases, and finally taking the sum of the output values.

In the following, we show that $-\varphi^{tot}(W)$ can be used as a surrogate regularizer for the $L_0$-norm to induce sparsity. Taking $R(W) = ||W||_{0,1}$ in Equation (1), it is easy to show that any neural network $W$ that minimizes $||W||_{0,1}$ while connecting the input layer to the output layer (without skip connections), i.e., $\varphi^{tot}(W) > 0$, has $K - 1$ non-zero weights. As the following theorem shows, a similar result holds for the CoNNect regularizer as any $W$ minimizing $-\varphi^{tot}(W)$ has between layer 2 and $K - 1$ only $K - 3$ non-zero weights.

**Theorem 1.** *Consider the problem*

$$\min_W -\varphi^{tot}(W), \tag{4}$$

*for a given network with number of layers $K > 2$. All solutions $W^*$ to Equation (4) have at most $|V_1| + |V_K| + K - 3$ non-zero weights.*

*Proof.* See Appendix A.1. □

Theorem 1 demonstrates that $L_0$-norm regularization can be effectively achieved through the CoNNect regularizer, as the induced sparsity in large neural networks is comparable. Importantly, the difference in the number of non-zero elements becomes negligible in practice when most input nodes contribute valuable predictive information, and all output nodes are used for accurate classification. Crucially, our regularizer does not force the input nodes to disconnect due to its indifference to the number of input nodes that connect to the second layer, which is a beneficial feature. If certain input nodes were disconnected, as might happen with other regularizers such as $L_1$-regularization, important data features could be disregarded, potentially resulting in suboptimal model performance.

We now show that a gradient descent can easily solve Equation (4). In the following, we assume that any network $W$ is connected, that is, $\varphi^{tot}(W) > 0$. We do so because we will prove later that it is impossible to reach an unconnected network ($\varphi^{tot}(W) = 0$) when starting in a connected network simply by using a log-transformation of $\varphi^{tot}(W)$.

First, consider for some $(i, j) \in E_k$ let

$$\partial_{W_{i,j}} (\theta(W))_{i,j} = \frac{\sum_{(r,c) \in E_k} |W_{r,c}| - |W_{i,j}|}{(\sum_{(r,c) \in E_k} |W_{r,c}|)^2}, \text{ and } \partial_{W_{q,t}} (\theta(W))_{i,j} = \frac{-|W_{q,t}|}{(\sum_{(r,c) \in E_k} |W_{r,c}|)^2},$$

specifically for $(q, t) \neq (i, j) \in E_k$. Observe that differentiating $\theta(W)$ with respect to $W_{i,j}$ only affects the weights in the same layer as $W_{i,j}$. Thus, a stationary point to Equation (4) solves the following first-order conditions:

$$\partial_{W_{i,j}} \varphi^{tot}(W) = \sum_{(r,c) \in E_1} \partial_{W_{r,c}} (\theta(W))_{i,j} \cdot a_{c\cdot} = 0, \quad \forall \, (i, j) \in E_1;$$

$$\partial_{W_{i,j}} \varphi^{tot}(W) = \sum_{(r,c) \in E_k} a_{\cdot r} \cdot \partial_{W_{r,c}} (\theta(W))_{i,j} \cdot a_{c\cdot} = 0, \quad k = 2, \ldots, K - 2, \forall \, (i, j) \in E_k;$$

$$\partial_{W_{i,j}} \varphi^{tot}(W) = \sum_{(r,c) \in E_{K-1}} a_{\cdot r} \cdot \partial_{W_{r,c}} (\theta(W))_{i,j} = 0, \quad \forall \, (i, j) \in E_{K-1}, \tag{5}$$

where $a_{\cdot r} = \sum_{i \in V_1} \sum_{\gamma \in \Gamma_{i,r}} \prod_{k=1}^{|\gamma|-1} (\theta(W))_{\gamma_k}$ and $a_{c\cdot} = \sum_{m \in V_K} \sum_{\gamma \in \Gamma_{c,m}} \prod_{k=1}^{|\gamma|-1} (\theta(W))_{\gamma_k}$ are the connectivity from input layer to a node $r$ and connectivity from a node $c$ to the output layer, respectively. To satisfy Equation (5), we need:

- the weights for the edges in $E_1$ must be assigned to all $(\theta(W))_{i,j}$, where $j \in \arg\max_p a_p$;

- the weights for the edges in $E_k$, $k = 2, \ldots, K-2$ must be assigned to $(\theta(W))_{i,j}$, where $(i,j) \in \arg\max_{(p,q)} a_{\cdot p} a_{q\cdot}$;

- the weights for the edges in $E_{K-1}$ must be assigned to $(\theta(W))_{i,j}$, where $i \in \arg\max_q a_{\cdot q}$.

Weight matrices $W$ that are local optima to Equation (4) can be characterized as having all paths between layers 2 and $K-1$ with equal strength, since stronger paths yield larger $\partial_{W_{i,j}} \varphi^{tot}(W)$ and so attract more weight; see Equation (5). Moreover, the paths need equivalent weights in the sequence as imbalances are inherently non-stationary. This insight implies that for all non-optimal stationary points, i.e., $\varphi^{tot}(W) < 1$, there exists a direction of improvement by simply transferring mass from one path to another. It follows that these solutions are inherently unstable and are not local optima. Concluding, all local optima to Equation (4) are global optima. We present the precise statement in the following theorem.

**Theorem 2.** *Assume a neural network with $K > 3$ layers. All stationary points $W^*$ to Equation (4) that are connected, i.e., $\varphi^{tot}(W) > 0$, have paths with equal subsequent weights between layers 2 and $K-1$ on its non-zero paths. That is, for each two paths $\gamma', \gamma'' \in \bigcup_{i \in V_1, m \in V_K} \Gamma_{i,m}$, such that*

$$\prod_{k=1}^{K-1} (\theta(W^*))_{\gamma_k} > 0, \quad \gamma \in \{\gamma', \gamma''\},$$

*i.e., both paths have positive weight, we have $(\theta(W^*))_{\gamma'_k} = (\theta(W^*))_{\gamma''_k}$, for all $k = 2, \ldots, K-2$. Moreover, the only stable stationary points $W^*$ of $-\varphi^{tot}(W)$ are global minimizers and so have only $K-3$ non-zero weights between layer 2 and $K-1$.*

*Proof.* See Appendix A.2. $\qquad\square$

As Theorem 2 shows, the only stable stationary points of CoNNect are those where the weight matrix does have between layer 2 and $K-1$ only $K-3$ non-zero weights. This implies that a gradient search algorithm will not get stuck in the other (unstable) stationary points as for those there is always a direction of improvement. Hence, global solutions to Equation (4) are easily found using a gradient search.

As argued earlier, it is recommended to take the logarithm over the connectivity regularizer, i.e.,

$$-\log\left(\varphi^{tot}(W)\right), \tag{6}$$

as it ensures that if the neural network tends to disconnect during training, i.e., $\varphi^{tot}(W) \to 0$, Equation (6) approaches $\infty$, hence preventing layer collapse. Moreover, it enhances numerical stability, ensuring that the regularization term remains well-behaved even for varying scales of connectivity.

**Implementation Details:** We provide our detailed implementation of CoNNect in Appendix B, spanning various modern neural network architectures. It is worth emphasizing that CoNNect is a highly flexible regularizer. While it is currently applied as detailed in Appendix B, its design allows for substantial flexibility in how it is applied. For instance, skip connections could be explicitly included in $W$, or CoNNect could be selectively applied to specific parts of a neural network. This selective application enables the regularization of targeted components of an architecture, ensuring connectivity improvements without interfering with the functionality of other parts of the network. Future work can explore this flexibility to further refine and adapt the CoNNect regularizer.

**Computational Efficiency:** As argued, $\varphi^{tot}(W)$ can be efficiently computed using a single forward pass (and its gradient with a single backward pass). Thus, for a batch size of $M$, the additional time used for computing $\varphi^{tot}(W)$ is proportional to $\frac{1}{M}$. Hence, CoNNect can be efficiently applied to large-scale neural networks without incurring significant computational overhead.

### 3.3.2 WEIGHT-LEVEL PRUNING

Once we have trained a model with CoNNect regularization, many of the redundant weights will have been pushed to zero. Consequently, we can hard prune the regularized model using pre-established pruning strategies. A well-known strategy is simple magnitude-based pruning (LeCun et al., 1989), which prunes the smallest weights in absolute value. Alternatively, we can use Syn-Flow pruning (Tanaka et al., 2020), which prunes the neural network's weights according to synaptic saliency scores:

$$I_{i,j} = \left(\partial_{(\theta(W))_{i,j}} \varphi^{tot}(W)\right) \cdot (\theta(W))_{i,j} = a_{\cdot i} \cdot (\theta(W))_{i,j} \cdot a_{j \cdot},$$

and eliminate the weights with the smallest $I_{i,j}$ values.

### 3.3.3 CHANNEL-LEVEL REGULARIZATION

The regularizer introduced in Section 3.3.1 was explicitly defined on the weights of the neural network, making it an unstructured pruning approach. In this section, we show how it can be easily extended to structured pruning. To this end, we can introduce a scaling factor for the output of structures (e.g., neurons, channels, etc.) that we want to prune (Huang & Wang, 2017). In the following, we explain how to include structured pruning on the channel-level in Convolutional Neural Networks (CNNs), but this can be naturally extended to any parallel structures in neural networks, such as nodes and entire block structures.

CNNs are a specialized type of neural network designed to process grid-like data such as images. These images can be represented using a tensor $X \in \mathbb{R}^{d \times h \times w}$, where $d$ refers to the number of channels (e.g., RGB for color images) and $h$ and $w$ refer to the height and width of the image respectively. A standard CNN consists of (several) convolutional layers followed by an activation function (e.g., ReLU), and pooling layers that reduce spatial dimensions while preserving important features. Convolutional layers transform the tensor into a set of feature maps through a series of learned filters (also known as kernels). Each convolutional layer in the CNN applies these filters to local regions of the input, capturing spatial hierarchies and patterns like edges, textures, and more complex shapes as the network deepens.

For performing regularizing on the channel-level, we introduce a set of learnable parameters that scale the output of each channel after a convolutional layer. More formally, for every $X^{(k)} \in \mathbb{R}^{d \times h \times w}$, which is the activation after the $k$-th convolutional layer, we scale the channels with $\delta^{(k)} \in \mathbb{R}^d$ so that

$$X^{(k)\prime} = \delta^{(k)} \odot X^{(k)},$$

where $\odot$ denotes element-wise multiplication so that the scaling factor $\delta^{(k)}$ is broadcast across the height $h$ and width $w$. The inclusion of scaling factors $\delta^{(k)}$ is a simple linear transformation and so can be perceived as the introduction of an additional layer to the neural network $W$, see Figure 2, resulting in an extended neural network denoted by $W'$. As the

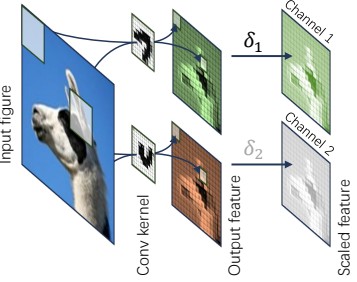

Figure 2: Illustration of CNN with the scaling factor.

normalization in Equation (2) will also be applied on the scaling factors, the unstructured CoNNect regularizer in Equation (3) carries over to a structured regularization, where the scaling factors of less informative channels are pushed to 0 and more informative channels are retained.

### 3.3.4 CHANNEL-LEVEL PRUNING

Once a regularized neural network is obtained, we can do pruning in a similar fashion as in Section 3.3.2. Specifically, we can prune its channels via calculating an importance scores for each channel. To that end, we aim to determine the contribution of a channel $c$ in layer $k$ in terms of the connectivity of the neural network, denoted by $I_{k,c}$. More formally, let $\theta_c^{(k)}(\delta) = |\delta_c^{(k)}|/\|\delta^{(k)}\|_1$ denote the normalization of the scaling factors with index $c$ for convolutional layer $k-1$ so that $I_{k,c}$ can be determined via

$$I_{k,c} = \left(\partial_{\theta_c^{(k)}(\delta)} \varphi^{tot}(W)\right) \cdot \theta_c^{(k)}(\delta) = \left(\sum_{r \in V_{k-1}^{(c)}} a_{\cdot r}\right) \cdot \theta_c^{(k)}(\delta) \cdot \left(\sum_{r \in V_{k+1}^{(c)}} a_{r \cdot}\right),$$

where $V_k^{(c)}$ is the subset of nodes in a layer $k$ corresponding to channel index $c$. Simply put, $I_{k,c}$ denotes the total connectivity that flows through channel $c$ in layer $k$. Consequently, a simple pruning strategy is to prune the channels with lowest values of $I_{k,c}$.

# 4 NUMERICAL EXPERIMENTS

## 4.1 WEIGHT-LEVEL PRUNING

In the following, we want to study the effects of integrating CoNNect regularization in an unstructured pruning task. Let us consider a small multilayer perceptron neural network with ReLU activations. The network has 6 input nodes, three hidden layers of 5 nodes, and a single output node. We sample input values $x_i = (x_{i,1}, \ldots, x_{i,6}) \sim \mathcal{N}(0, \Sigma)$, where $\Sigma$ is a matrix with the value 2 on the diagonal. Furthermore, we let the output values be

$$y_i = \begin{cases} 1 & \text{if } x_{i,1} + x_{i,2} + \xi_i > 0; \\ 0 & \text{otherwise,} \end{cases}$$

where $\xi_i \sim \mathcal{N}(0, 0.25)$. To find a sparse network representation, we train the network with $L_1$ and CoNNect regularization, and remove the unimportant weights after training. To that end, we solve

$$\min_{W,b} \quad \mathcal{L}(\hat{y}, y) + \lambda_1 \|W\|_{1,1} - \lambda_2 \log\left(\varphi^{tot}(W)\right) + \lambda_3 \|W\|_{2,1}, \tag{7}$$

where $\mathcal{L}(\hat{y}, y)$ is the Binary Cross Entropy between target and input probabilities ,and $\|W\|_{2,1}$ is the often-applied $L_2$-regularization (weight decay). We fit three different models following Equation (7), for which we provide coefficients in Table 1. All models have been trained for 200 epochs using Adam with a learning rate of 0.01, a cosine annealing scheduler, and batch size 256. After training, we pruned 96% of the weights in each layer using the pruning strategies discussed in Section 3.3.2: i) magnitude pruning, and ii) SynFlow pruning. Finally, the model is fine-tuned with the same hyperparameters but with a decreased initial learning rate of 0.001 for 50 epochs.

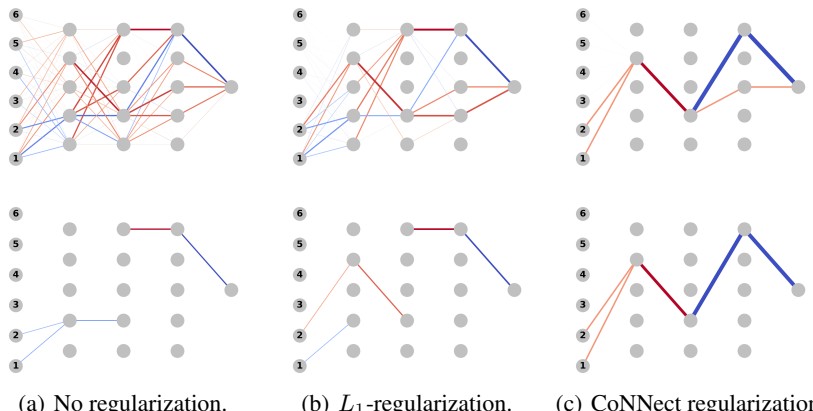

(a) No regularization.    (b) $L_1$-regularization.    (c) CoNNect regularization.

Figure 3: Trained (top) and fine-tuned (bottom) models. Thicker and darker colors correspond to stronger values. Red and blue edges correspond to positive and negative values respectively.

We show the results in Figure 3 for a single neural network initialization with SynFlow pruning. We present the results for 100 repetitions, where we show the (aggregated) train and test loss in Figures 4(a) and (b) in and the fine-tuned accuracies in Figure 4(c) and (d). Roughly speaking, the final accuracy for each model can be categorized by the ability to find the network connecting the

Table 1: Regularizer coefficients.

| Regularizer | $\lambda_1$ | $\lambda_2$ | $\lambda_3$ |
|---|---|---|---|
| None | 0 | 0 | 0.0005 |
| $L_1$ | 0.001 | 0 | 0.0005 |
| CoNNect | 0 | 0.1 | 0.0005 |

input nodes 1 and 2 to the output layer. If the fine-tuned accuracy is around $0.50$, the algorithm was unable to connect node 1 and node 2 to the output (e.g., see Figures 3(a) and (b)). If the fine-tuned accuracy is around $0.75$, the algorithm was able to connect node 1 or node 2 to the output. Finally,

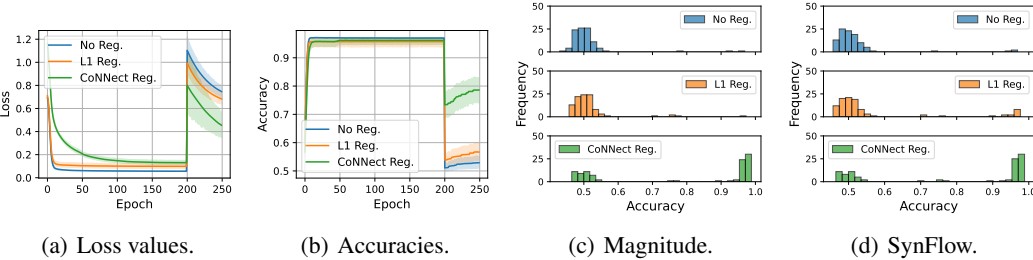

(a) Loss values. (b) Accuracies. (c) Magnitude. (d) SynFlow.

Figure 4: (a)-(b) Learning curves for solving Equation (4). Synflow pruning happens at iteration 200. Bandwidths are 95% confidence intervals. (c)-(d) Fine-tuned accuracy after magnitude pruning and SynFlow pruning of regularized models.

if the algorithm preserved the edges connecting node 1 and node 2, it found the correct network and achieves an accuracy of more than $0.95$ (e.g., see Figure 3(c)).

As shown in Figure 4(c) and (d), CoNNect regularization via $\varphi^{tot}(W)$ is beneficial to both pruning strategies. It is noteworthy that SynFlow pruning does not offer any further improvement over connectivity regularization compared to simple magnitude pruning. This can be attributed to the fact that CoNNect regularization has already trained the network to use the correct paths to model the current problem, as shown in Figure 3(c). It thus suffices to apply a simple magnitude pruning to identify these paths.

**Remark:** Synflow is traditionally introduced as a pre-training pruning method, its data-agnostic nature makes it less effective in this context, given the presence of uninformative input nodes. Moreover, SynFlow is generally regarded as a global pruning strategy. However, we frequently observed layer collapse under this configuration. In contrast, applying a local pruning approach yielded significantly better results, particularly for models without regularization and $L_1$ regularization. We thus show the results using a local pruning approach.

To show the robustness of our results, we conduct an ablation study to analyze the impact of the regularization strengths, see Appendix D.1.

### 4.2 CHANNEL-LEVEL PRUNING

In this section, we demonstrate CoNNect for structured pruning on the channel-level; see Section 3.3.4. To that end, we train VGG-11 (Simonyan & Zisserman, 2014) (including Batch Normalization (BN) layers) on the CIFAR-10 (Krizhevsky et al., 2009) dataset. Since the BN-layers have weights that scale channels in VGG-11 independently, it suffices to use these as scaling factors. Hence, there is no need to introduce another set of scalars for scaling the channel output.

For the regularization of connectivity in VGG-11, two things are worth mentioning. First, the standardization applied in BN layers can be disregarded, as it merely re-scales the connectivity values at these nodes. Second, we remove dropout layers, as they do not contribute to neural network connectivity. Third, we replaced the max pooling layers with average pooling layers to ensure that all paths contribute consistently throughout the network and for numerical stability. Note these changes are

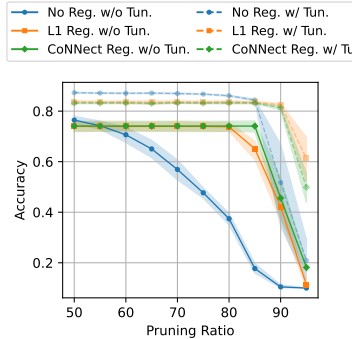

Figure 5: Accuracy for given pruning ratio. The shadow demonstrate $98\%$ confidence intervals.

only implemented when computing the forward pass for CoNNect, the forward pass for the VGG-11 itself is not modified. Similar to Section 4.1, we train VGG-11 via Equation (7) and compare the results for: i) no regularization, ii) $L_1$ regularization, and iii) CoNNect regularization. We train various models for 20 epochs with parameters shown in Table 3, Appendix D.1, and fine-tune the model after pruning (see Section 3.3.4) for 5 epochs each. The results, presented in Figure 5, are obtained by 10 repeats and similar to Section 4.1 show CoNNect can outperform $L_1$ and $L_2$ regularization. For further evaluation on GNNs, please see Appendix D.2.

### 4.3 ONE-SHOT PRUNING LLMS VIA CONNECT

To further demonstrate the versatility and scalability of CoNNect, we integrate it in the framework of LLM-Pruner (Ma et al., 2023) to perform a one-shot pruning on LLaMA-7B (Touvron et al., 2023). First, all parameters of the LLM are divided into several groups according to the dependency relationships in the computation process. Then, the importance score under the objective function $\mathcal{J}(\cdot)$ is calculated by $I_{i,j} = |\mathcal{J}_{W_{i,j}}(W) - \mathcal{J}_{W_{i,j}=0}(W)| \approx |\partial_{W_{i,j}} \mathcal{J}(W) \cdot W_{i,j}|$, where we redefine $(\theta(W))_{i,j} = |W_{i,j}|$ to enhance both numerical stability and computational efficiency, as dropping the normalization does not affect the ranking of importance scores or the outcomes. We integrate our CoNNect approach to the LLM-Pruner through the objective, i.e., $\mathcal{J}(W) = \mathcal{L}(\mathcal{D}) - \lambda \log(\varphi^{tot}(W))$, where $\mathcal{D}$ denotes the dataset. The importance of each group is aggregated through summation, and the least important groups are pruned. Finally, the LLM is fine-tuned using the LoRA (Hu et al., 2021) technique to restore as much of the maximum structural capability as possible under the current architecture.

To assess the model performance, we conduct a zero-shot perplexity analysis on WikiText2 (Merity et al., 2022) and PTB (Marcus et al., 1993), and then follow Gao et al. (2021) to test the model with zero-shot classification tasks on common sense reasoning datasets: BoolQ (Clark et al., 2019), PIQA (Bisk et al., 2020), HellaSwag (Zellers et al., 2019), WinoGrande (Sakaguchi et al., 2021), ARC-easy, ARC-challenge (Clark et al., 2018), OpenbookQA (Mihaylov et al., 2018), where the model ranks the choices in these multiple-choice tasks.

We compare CoNNect to $L_2$, random, and vanilla LLM-Pruner's importance metrics with a $40\%$ parameter reduction. All methods are equipped with the same group division and aggregation strategy. As presented in Table 2, compared to vanilla LLM-Pruner, we have reduced the performance gap between the pruned model and the original model by $9.13\%$ without fine-tuning, which is $9.29\%$ when fine-tuning is applied. Essentially, CoNNect enhances the LLM-Pruner's framework with an extra consideration of connectivity, providing good results. The results differ significantly from those obtained by randomly removing parameter groups, but the grouping approach keeps random pruning from detrimental outcomes. However, $L_2$ regularization even results in incorrect pruning choices, which is consistent with the conclusions drawn in the previous two subsections. Please refer to Appendix C.2 for detailed experimental settings and Appendix D.3 for more evaluation aspects.

Table 2: Zero-shot performance of the compressed LLaMA-7B. High scores are better, except for WikiText2 and PTB (indicated by the downward arrow). The bold values indicate the best results. The average is calculated among seven classification accuracies. An asterisk denotes that performance normalization is not available.

| Pruned Model | Method | WikiText2↓ | PTB↓ | BoolQ* | PIQA | HellaSwag | WinoGrande* | ARC-e | ARC-c | OBQA | Average |
|---|---|---|---|---|---|---|---|---|---|---|---|
| Ratio = 0% | LlaMA-7B | 12.62 | 22.15 | 73.15 | 77.48 | 73.01 | 67.09 | 52.57 | 41.47 | 42.40 | 61.02 |
| Ratio = 40% w/o tune | L2 | 13783.81 | 27844.06 | 42.69 | 52.01 | 28.29 | 51.46 | 27.36 | 25.85 | 29.80 | 36.78 |
| | Random | 100.42 | 133.56 | 40.00 | 57.29 | 36.00 | 50.12 | 32.83 | 25.77 | 31.00 | 39.00 |
| | LLM-Pruner | 48.09 | 105.24 | 58.90 | 64.74 | 47.58 | **53.20** | 37.75 | 29.44 | 35.00 | 46.66 |
| | CoNNect | **46.43** | **95.08** | **60.95** | **67.30** | **50.04** | 52.09 | **38.30** | **29.86** | **36.80** | **47.91** |
| Ratio = 40% w/ tune | L2 | 44.91 | 67.16 | 47.34 | **71.60** | 50.60 | 54.38 | 43.35 | 32.25 | 36.80 | 48.05 |
| | Random | 37.82 | 58.12 | 54.95 | 67.36 | 48.61 | 55.25 | 43.69 | 30.29 | 33.20 | 47.62 |
| | LLM-Pruner | 27.62 | 48.28 | 59.97 | 71.38 | 56.21 | **59.35** | 44.53 | 32.42 | 36.20 | 51.44 |
| | CoNNect | **27.13** | **47.44** | **61.59** | 71.06 | **57.78** | 58.48 | **45.58** | **32.85** | **39.00** | **52.33** |

## 5 CONCLUSIONS

In this work, we introduce a novel regularizer called CoNNect, which leverages network connectivity to promote sparsity. Theoretically, we showed that CoNNect aligns with the minimization of the $L_0$-norm and avoids getting trapped in local minima. Through numerical experiments, we have shown that CoNNect can be effectively applied in many pruning strategies. Moreover, it can be used for both unstructured and structured network pruning. Specifically, we showed how CoNNect as regularizer improves the pruning of MLPs, CNNs and GNNs, compared with standard $L_1$-regularization. Furthermore, we demonstrated how CoNNect can be applied competitively in a one-shot pruning framework for large language models (LLMs), as proposed by Ma et al. (2023). This shows that CoNNect offers flexibility in its implementation within different pruning strategies.

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

## A  PROOFS

### A.1  PROOF THEOREM 1

Let $\Gamma_{i,m}$ denote the set of paths in the neural network that go from some input node $i \in V_1$ to the output node $m \in V_K$, where

$$\gamma = ((i,j), (j,k), \ldots, (l,m)) \in \Gamma_{i,m}$$

is a sequence of edges from the input layer to the output layer. Using that $\varphi^{tot}(W)$ is the sum of weights of paths from the input to the output layer (Neyshabur et al., 2015), we rewrite

$$\varphi^{tot}(W) = \sum_{i \in V_1} \sum_{m \in V_K} \sum_{\gamma \in \Gamma_{i,m}} \prod_{k=1}^{K-1} (\theta(W))_{\gamma_k} = \sum_{i \in V_1} \sum_{m \in V_K} \sum_{\gamma \in \Gamma_{i,m}} \prod_{k=1}^{K-1} \frac{|W_{\gamma_k}|}{\sum_{(r,c) \in E_k} |W_{r,c}|},$$

where $\gamma_k$ refers to the $k$th edge in a sequence $\gamma$. Then, to minimize $R(W)$, i.e., maximize $\varphi^{tot}(W)$, we need to allocate all the mass to a single path from the input to the output, which means selecting a specific sequence of weights that maximizes the product along that path, effectively minimizing the contributions from all other paths.

To show the upper bound of $|V_1| + |V_K| + K - 3$ non-zero weights in $W^*$, assume w.l.o.g. some $W^*$ where a single path $\Gamma_{i,m}$ has all mass in the network. It follows that $\varphi^{tot}(W^*) = 1$. Now, let $W'$ denote a solution where some mass from the first weight $W_{i,j}$, for $(i,j) \in \Gamma_{i,m}$ is shifted to any other weight(s) $W_{l,j}$ (note that $j$ is fixed), where $l \in V_1$ connects to $j \in V_2$. It is easily seen that $\varphi^{tot}(W') = 1$ since

$$\varphi^{tot}(W') = \sum_{l \in V_1} (\theta(W'))_{l,j} \sum_{\gamma \in \Gamma_{j,m}} \prod_{k=1}^{K-1} (\theta(W'))_{\gamma_k}$$

$$= \sum_{l \in V_1} \frac{|W'_{l,j}|}{\sum_{(r,c) \in E_1} |W'_{r,c}|} \sum_{\gamma \in \Gamma_{j,m}} \prod_{k=1}^{K-1} (\theta(W'))_{\gamma_k} = \sum_{l \in V_1} \frac{|W'_{l,j}|}{\sum_{(r,c) \in E_1} |W'_{r,c}|} \cdot 1 = 1,$$

In words, $\varphi^{tot}(W)$ is indifferent in how many of the $|V_1|$ input nodes connect to a single node in the second layer. Note that a similar argument can be made for the weights connecting the $K - 1$th layer with the $K$th layer. It follows that the number of non-zero weights for $W^*$ is upper bounded by $|V_1|$ for the first layer, $|V_K|$ for layer $K - 1$, and $K - 3$ for the weights of the remaining layers. The resulting upper bound is then $|V_1| + |V_K| + K - 3$.

### A.2  PROOF THEOREM 2

We prove this by induction using the necessary and sufficient system of equations for stationarity in $\varphi^{tot}(W)$, see Equation (5). Assume any connected neural network, i.e., $\varphi^{tot}(W) > 0$, of arbitrary size with $K = 2$ layers and weight allocation such that $(\theta(W))_{i,j} > 0$ for $i \in V_1$ and $j \in \arg\max_{k \in V_2} a_{.k}$. Note that for this specific case any weight allocation will be stationary in $\varphi^{tot}(W)$. Moreover, assume $a_{.i} = a_{.j}$, for all $i, j \in \arg\max_{k \in V_2} a_{.k}$, since adding a layer $V_{K+1}$ implies that this condition must hold to satisfy Equation (5) in the next step.

Now we add a new layer of arbitrary size $V_{K+1}$. In case $V_{K+1}$ is the last layer, it is sufficient to allocate $(\theta(W))_{i,j} > 0$, for all $i \in \arg\max_{k \in V_K} a_{.k}$ to obtain a stationary point. In case the neural network is expanded with another layer $V_{K+2}$ in a next step, we let $(\theta(W))_{i,j} > 0$ for $i \in \arg\max_{k \in V_K}$ and $j \in \arg\max_{k \in V_{K+1}} a_{.k}$, such that $a_{.i} = a_{.j}$, for all $i, j \in \arg\max_{k \in V_{K+1}} a_{.k}$ to satisfy Equation (5). Note that this immediately implies $(\theta(W))_{i,j} = (\theta(W))_{r,c}$, for all $(i,j), (r,c) \in \arg\max_{(i,j) \in E_{K+1}} a_{.i}a_{.j}$. Hence, $(\theta(W))_{\gamma'_k} = (\theta(W))_{\gamma''_k}$, for all $k = 2, \ldots, K - 2$, for all paths $\gamma$ with positive path weight. Moreover, note that stationarity cannot be induced by reparameterization $\theta(W)$. Considering that we derived the above points using the necessary and sufficient conditions for stationarity, all other points are non-stationary. Moreover, for all non-optimal stationary points, i.e., $\varphi^{tot}(W) < 1$, there exists a direction of improvement by simply transferring mass from one path to another. It follows that these solutions are inherently unstable and are not local optima. Hence, all local optima to Equation (4) are global optima.

## B  IMPLEMENTATION DETAILS

In this section, we outline how $\varphi^{tot}(W)$ can be efficiently computed using a slightly modified forward pass of the neural network and a vector of ones as input. Below, we outline how different modules are treated in this modified forward pass. The ability to handle these modules enables the application of CoNNect across a broad spectrum of neural network architectures.

**Linear Layers:** This includes both dense (fully connected) layers and convolutional layers. The weights of these layers define the primary connections between nodes and we normalize their weights via Equation (2). The biases, however, merely shift activations (which we will exclude), and do not influence connectivity structure and are therefore excluded.

**BN Layers:** Batch normalization layers apply standardization and scaling to the outputs of preceding layers. For the purposes of connectivity analysis, the standardization can be disregarded as it does not alter the structure of connections, but rather rescales values. Thus, we consider BN layers as identity mappings with preserved connectivity.

**Activation Functions:** Non-linear activation functions such as ReLU, sigmoid, or tanh are ignored. These functions transform node outputs but do not influence the underlying connectivity. Ignoring them simplifies the analysis without affecting the structural representation.

**Pooling Layers:** Max-pooling layers are replaced with average pooling layers. This change ensures that all input connections are treated equally in the computation of connectivity, rather than prioritizing the strongest signal as in max-pooling.

**Dropout:** Dropout layers are designed to randomly disable connections during training as a regularization method. Since they are stochastic and transient, they are ignored for connectivity analysis, as they do not represent fixed structural relationships.

**Identity Connections:** Identity connections, such as skip connections in residual networks, can be included when computing connectivity. However, since these connections (generally) are not parameterized, they can be ignored when optimizing the neural network's connectivity. Thus, we omit the identity connection in the forward pass.

## C  EXPERIMENTAL SETTINGS

**Platform**: All experiments were performed on a single NVIDIA RTX4090 GPU with 24GB of memory.

### C.1  EXPERIMENTAL SETTINGS FOR SECTION 4.2

**Dataset:** We use CIFAR-10 (Krizhevsky et al., 2009), a dataset with 60,000 32x32 images with 10 different classes. Each class has 6,000 images.

**VGG-11 (with Batch Normalization):** The VGG-11 model consists of 11 layers with learnable parameters, including 8 convolutional layers. Each convolutional layer is followed by a batch normalization (BN) layer and a ReLU activation function. Max pooling (2x2, stride 2) is applied where applicable, based on the spatial dimensions. The network concludes with a classifier composed of 3 fully connected (linear) layers.

Table 3: Regularizer coefficients used in Section 4.2.

| Regularizer | $\lambda_1$ | $\lambda_2$ | $\lambda_3$ |
|---|---|---|---|
| None | 0 | 0 | 0.001 |
| $L_1$ | 0.0001 | 0 | 0.001 |
| CoNNect | 0 | 0.1 | 0.001 |

## C.2 EXPERIMENTAL SETTINGS FOR SECTION 4.3

In the current experiment, we use 10 randomly selected samples from Bookcorpus (Zhu, 2015) to be the calibration samples for establishing the dependency between parameters in the model and calculate the gradient for LLaMA-7B. To that end, we truncate each sample to a sequence length of 128. We set the coefficient $\lambda$ of connectivity metric as $1 \times 10^5$. During fine-tuning, we utilize Alpaca (Taori et al., 2023), which comprises approximately 50,000 samples, to recover the capacity of the pruned model, which requires just 2 hours on our platform (NVIDIA RTX4090 GPU).

To determine which groups to prune, we compute importance scores for each weight in the model. Since the simplified form $(\theta(W))_{i,j} = |W_{i,j}|$ works well on LLMs, we use absolute values instead of normalized ones to reduce computational effort. We approximate connectivity by keeping all modules as they are for efficient and ease of implementation. Moreover, to evaluate connectivity, we use multiple inputs uniformly sampled between 0 and the vocabulary size instead of a single all-one. This is to avoid conflicts with reserved input values during the training process, leading to improper connectivity evaluation. Then, specifically for $L_2$ pruning, we compute the importance of each group by computing the $L_2$-norm and prune the groups with lowest importance scores. For random pruning, there is no need to compute importance scores for each group - we simply randomly select certain groups for pruning. Moreover, we leave the first three layers and the final layer unchanged (similar to Ma et al. (2023)), as substantial changes to the parameters of these layers greatly influence the performance of the model. Finally, the discovered groups within each module are pruned according to a predetermined ratio. The pruning rate for the selected groups is higher than the pruning ratio for the parameters since some layers (e.g., the excluded layers) retain their parameters. For a total of $40\%$ parameter removal, we must prune $50\%$ of the groups specifically from the fourth to the thirtieth layer.

# D SUPPLEMENTAL EXPERIMENTS

## D.1 ABLATION STUDY OF UNSTRUCTURED PRUNING ON MLPs

We have performed experiments with different values of $\lambda$. Specifically, increasing $\lambda_1$ by one order of magnitude to 0.01 causes a frequent occurrence of layer collapse, although it does increase the performances for the cases without layer collapse, see Figure 6 in Appendix D.1. Changing $\lambda_2$ by one order of magnitude to 1 did not cause any specific change, arguing for the stability of CoN-Nect. Moreover, increasing $\lambda_3$ by one order of magnitude to 0.005 seems to improve the model performance overall, especially for the CoNNect regularized model, see Figure 7 in Appendix D.1. Increasing $\lambda_3$ by another order of magnitude still shows very competitive results for CoNNect. Finally, we decrease $\lambda_1$ and $\lambda_2$ to 0.0005 and 0.05 respectively, and see that the regularizers become too weak leading the results to converge toward those of standard $L_2$ regularization.

Table 4: Regularizer coefficients used for producing Figure 6.

| Regularizer | $\lambda_1$ | $\lambda_2$ | $\lambda_3$ |
|---|---|---|---|
| None | 0 | 0 | 0.0005 |
| $L_1$ | 0.01 | 0 | 0.0005 |
| CoNNect | 0 | 1.0 | 0.0005 |

Table 5: Regularizer coefficients used for producing Figure 7.

| Regularizer | $\lambda_1$ | $\lambda_2$ | $\lambda_3$ |
|---|---|---|---|
| None | 0 | 0 | 0.005 |
| $L_1$ | 0.001 | 0 | 0.005 |
| CoNNect | 0 | 0.1 | 0.005 |

Table 6: Regularizer coefficients used for producing Figure 8.

| Regularizer | $\lambda_1$ | $\lambda_2$ | $\lambda_3$ |
|---|---|---|---|
| None | 0 | 0 | 0.05 |
| $L_1$ | 0.001 | 0 | 0.05 |
| CoNNect | 0 | 0.1 | 0.05 |

Table 7: Regularizer coefficients used for producing Figure 9.

| Regularizer | $\lambda_1$ | $\lambda_2$ | $\lambda_3$ |
|---|---|---|---|
| None | 0 | 0 | 0.005 |
| $L_1$ | 0.0005 | 0 | 0.005 |
| CoNNect | 0 | 0.05 | 0.005 |

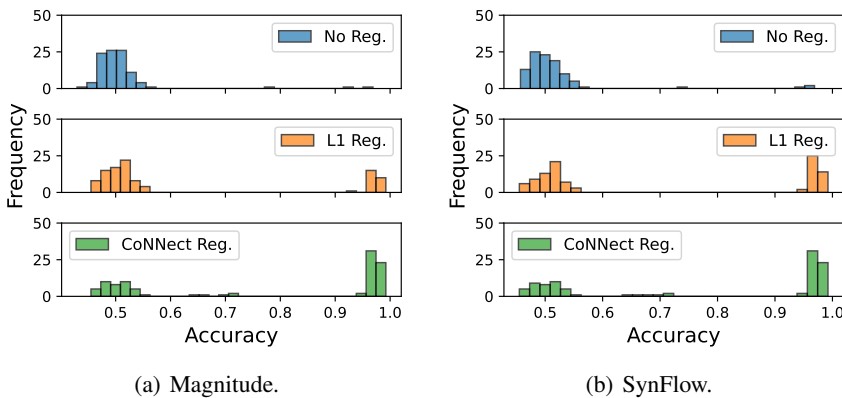

(a) Magnitude.

(b) SynFlow.

Figure 6: Fine-tuned accuracy after magnitude pruning and SynFlow pruning for the parameters in Table 4.

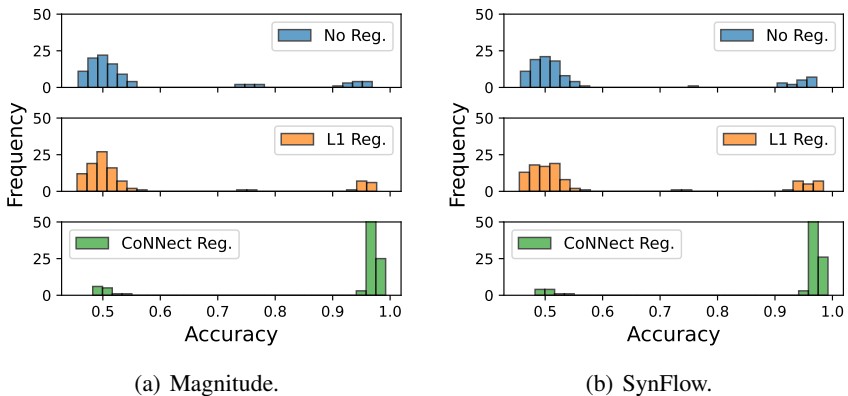

(a) Magnitude.

(b) SynFlow.

Figure 7: Fine-tuned accuracy after magnitude pruning and SynFlow pruning for the parameters in Table 5.

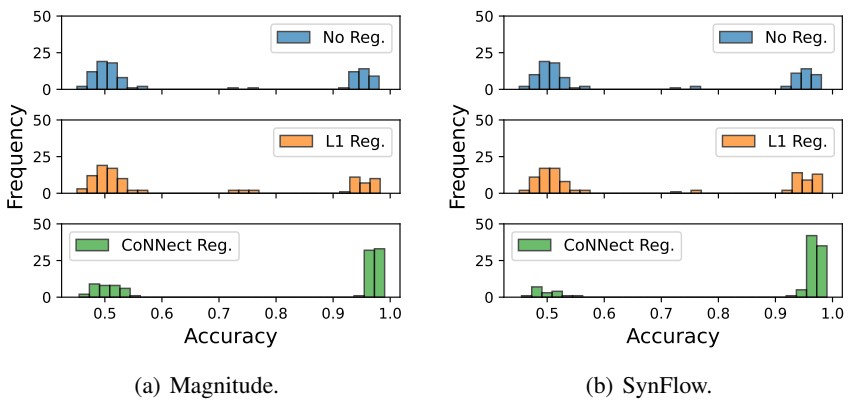

(a) Magnitude.

(b) SynFlow.

Figure 8: Fine-tuned accuracy after magnitude pruning and SynFlow pruning for the parameters in Table 6.

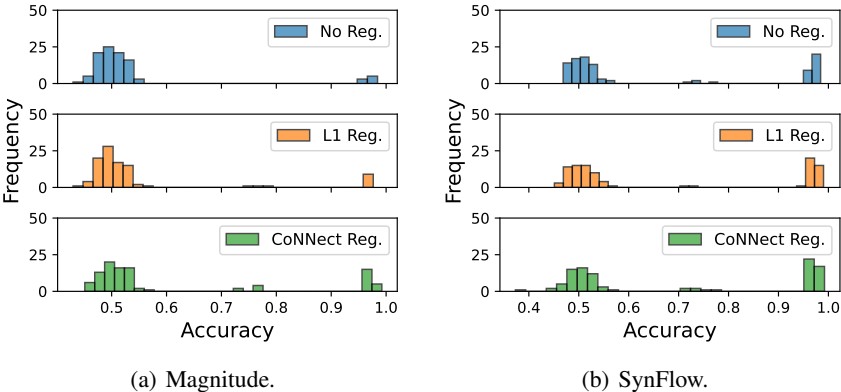

(a) Magnitude.

(b) SynFlow.

Figure 9: Fine-tuned accuracy after magnitude pruning and SynFlow pruning for the parameters in Table 7.

## D.2 CHANNEL-LEVEL PRUNING ON GRAPH NEURAL NETWORKS

Analogously to Section 4.2, we conduct extra experiments of pruning Graph Convolutional Network (GCN, Kipf & Welling, 2016) on Cora (Sen et al., 2008) dataset. Cora is a graph-based dataset consisting of 2,708 academic papers (nodes) and 5,429 citation links (edges), with each paper categorized into one of seven topics and represented by a 1,433-dimensional binary feature vector. And the GCN consists of 7 layers with learnable parameters, where the hidden feature dimensions are 512-256-256-256-256-64. Each GCN layer is followed by a ReLU activation function.

We train GCNs for 100 epochs using the parameters shown in Table 8 and fine-tune each model after pruning for 20 epochs. We conduct 10 repeated experiments, and as shown in Figure 10, our method consistently outperforms $L_1$-regularization. The shaded regions represent 98% confidence intervals.

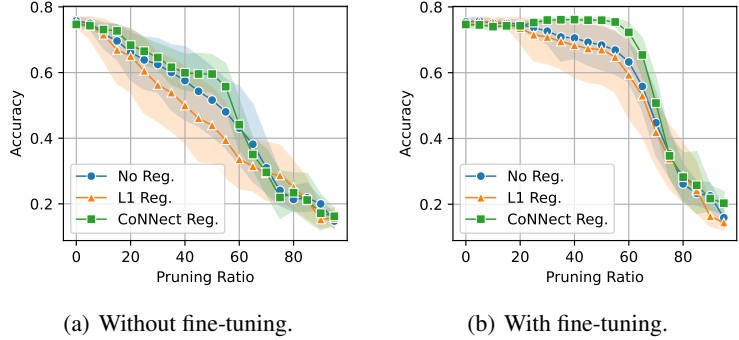

(a) Without fine-tuning.

(b) With fine-tuning.

Figure 10: Accuracies of GNNs for given pruning ratios.

Table 8: Regularizer coefficients used in GNN pruning.

| Regularizer | $\lambda_1$ | $\lambda_2$ | $\lambda_3$ |
|---|---|---|---|
| None | 0 | 0 | $1 \times 10^{-5}$ |
| $L_1$ | $1 \times 10^{-6}$ | 0 | $1 \times 10^{-5}$ |
| CoNNect | 0 | 0.1 | $1 \times 10^{-5}$ |

## D.3 STRUCTURAL PRUNING ON CNNs

Similarly to Section 4.3, we also integrate CoNNect into the preceding work of LLMPruner, Dep-Graph (Fang et al., 2023), to perform structural pruning on ResNet-56 (He et al., 2016) and VGG-19 (Simonyan & Zisserman, 2015), which are pretrained and fine-tuned on CIFAR-10 and CIFAR-100 datasets (Krizhevsky et al., 2009), respectively. DepGraph framework iteratively prunes the model until the predefined speed-up targets are achieved, which is calculated as the ratio of multiply-accumulate operations before and after pruning. We first follow the pruning intensity tested in Fang et al. (2023) and then verify CoNNect with extreme cases. Thus, the pruning is set to target speed-ups of $2.5\times$ and $16\times$ for ResNet-56 on CIFAR-10 and $8\times$ and $16\times$ for VGG-19 on CIFAR-100. As shown in Table 9, CoNNect exhibits advantages across various pruning ratios, with the benefits being more pronounced in more extreme cases.

Table 9: Results of CNN pruning under different settings.

| Model & Dataset | Base Acc. | Method | Pruned Acc. | Speed Up | Pruning Ratios |
|---|---|---|---|---|---|
| ResNet-56 & CIFAR-10 | 93.53 | DepGraph | 93.17 | $2.51\times$ | 56.22 |
| | | CoNNect | **93.63** | $2.50\times$ | 53.20 |
| | | DepGraph | 80.24 | $16.17\times$ | 98.27 |
| | | CoNNect | **83.12** | $17.24\times$ | 97.46 |
| VGG-19 & CIFAR-100 | 73.50 | DepGraph | 65.89 | $8.12\times$ | 90.48 |
| | | CoNNect | **69.38** | $8.00\times$ | 93.33 |
| | | DepGraph | 57.48 | $16.10\times$ | 96.14 |
| | | CoNNect | **62.56** | $16.07\times$ | 97.51 |

## E  LIMITATIONS AND FUTURE WORK

For future work, we suggest extending CoNNect to incorporate biases and activation functions, thereby broadening its applicability. Moreover, when CoNNect is applied as regularizer, it can be used to determine meaningful pruning ratios by analyzing the dispersion achieved in the model's regularized weights (e.g., see Zhuang et al. (2020)), where in the current paper's experiments we have simply used a predetermined pruning ratio. Finally, exploring its effectiveness on different neural network architectures, such as recurrent neural networks, could provide further insights into its generalizability.

