# OpenReview forum: "CoNNect: A Swiss-Army-Knife Regularizer for Pruning of Neural Networks"
_ICLR.cc/2025/Conference — Submitted to ICLR 2025_

### Official Review · Reviewer_DVNq · 2024-10-21

**Soundness:** 2
**Presentation:** 2
**Contribution:** 2
**Rating:** 3
**Confidence:** 3

**Summary:**

The paper introduces a novel differentiable regularizer, CoNNect, for training sparse neural networks. Inspired by Katz centrality from graph theory, CoNNect is designed to enhance neural network pruning techniques while maintaining key connectivity between the input and output layers throughout training. Traditional pruning methods often rely on L1-regularization as a surrogate for L0-norm, but these methods may lead to issues like layer collapse and loss of network structure. CoNNect addresses these limitations by ensuring the network retains essential paths, thereby preserving the model’s performance during pruning.

The paper demonstrates CoNNect's ability to approximate L0-regularization more effectively than L1-regularization, leading to improved sparsity without sacrificing connectivity. It is suitable for both unstructured and structured pruning techniques. The authors validate CoNNect's performance across several experiments, including channel-level pruning in CNNs and one-shot pruning for large language models, demonstrating improved performance compared to existing pruning techniques.

**Strengths:**

1. **Originality**:
   The paper introduces an original approach by leveraging Katz centrality, a concept from graph theory, to measure and preserve neural network connectivity during pruning.

2. **Quality**:
The authors establish the stability and effectiveness of CoNNect in approximating L0 regularization while avoiding layer collapse.

3. **Clarity**:
   The paper is fairly well-written and structured.

4. **Significance**:
The CoNNect regularizer directly tackles the limitations of L1 regularization and other sparsity-inducing methods, which often lead to disconnected or underperforming models. By maintaining network connectivity, CoNNect ensures that pruned models retain their predictive power and functional integrity.

**Weaknesses:**

1. **Complexity of Implementation**:
   The complexity of implementing the CoNNect regularizer in practice. The reliance on Katz centrality, which is not typically a standard tool in neural network training, introduces additional computational overhead. Furthermore, the paper does not provide sufficient guidelines or discussion about how CoNNect scales with larger and more complex neural network architectures, particularly in real-world environments.

2. **Lack of Comprehensive Comparisons**:
   While the paper demonstrates that CoNNect outperforms traditional L1 and L2 regularization and some pruning methods like SynFlow, it lacks a comparison with other contemporary pruning techniques. For instance, techniques like Lottery Ticket Hypothesis-based pruning (Frankle & Carbin, 2019), DLTH  (Bai et al. 2022),  or other structured pruning methods using adaptive sparsity (e.g., Movement Pruning by Sanh et al., 2020) are notably absent. To strengthen the empirical validation, the authors should include experiments that compare CoNNect with these other pruning frameworks.

3. **Limited Architectural Diversity in Experiments**:
   The paper focuses mainly on feedforward networks, convolutional neural networks (CNNs), and large language models (LLMs), but does not include experiments on other widely-used architectures such as recurrent neural networks (RNNs), transformers, or graph neural networks (GNNs). Since the authors state that CoNNect can be applied across various neural network architectures, demonstrating its effectiveness on a broader range of models would help substantiate this claim.

4. **Insufficient Exploration of Hyperparameters**:
   The paper uses fixed hyperparameter values for CoNNect across different experiments, but there is limited discussion on how sensitive CoNNect is to these hyperparameters or how they should be tuned for different models or datasets. Given that regularization methods can be highly sensitive to parameter tuning, especially in complex neural networks, it would be beneficial for the authors to conduct an ablation study or sensitivity analysis on the regularization coefficients used in CoNNect.

**Questions:**

I would like to see more comprehensive comparisons with other recent pruning techniques for large language models (LLMs). For instance, the following two papers report higher accuracies at more aggressive levels of pruning, as evidenced by Table 23 in the *Wanda* paper (the second in the list):

1. **SparseGPT: Massive language models can be accurately pruned in one-shot** (Elias Frantar and Dan Alistarh, ICML 2023)
2. **A simple and effective pruning approach for large language models** (Sun et al., ICLR 2024)

Incorporating these methods into the experimental results would provide a clearer picture of how **CoNNect** performs relative to state-of-the-art pruning techniques for LLMs, particularly when it comes to accuracy and the extent of pruning.

Additionally, the experiments in the current paper are not sufficiently comprehensive. The results are reported for only one LLM architecture and one model size. However, in empirical studies of this nature, it is common practice to evaluate multiple model sizes across different architectures to ensure generalizability. Expanding the range of experiments would help substantiate the claims made about CoNNect’s effectiveness and scalability.

---

> ### Author Response · Authors · 2024-11-23
> **Response to Reviewer DVNq (Part 1)**
>
> **[RW1]:** We understand the reviewer's concern about the complexity of the implementation of CoNNect. The computational overhead is in fact very moderate given that CoNNect can be computed in a time that is bounded by the time it takes for a single forward pass the neural network. More formally, for a batch size of $M$, the time used for computing CoNNect is proportional to $1/M$. We kindly refer the reviewer to our response [RQ1] of reviewer SZg4 and [RQ1] of eRZy. We have emphasized this point in the revised manuscript in lines 310-314.
>
> **[RW2]:** We understand the reviewer's concern. However, it is not our goal to compare CoNNect with the suggested pruning frameworks, such as Lottery Ticket Hypothesis-based pruning (Frankle & Carbin, 2018). Instead, we want to demonstrate how CoNNect can be *integrated* into such pruning frameworks. We believe CoNNect is a conceptual contribution that, without much effort, can be used to enhance many hard pruning approaches (as we show for magnitude pruning and LLM-pruner). The fact that we can show good results argues for the importance of connectivity in neural network performance (Axiom 2). In the paper, we show how magnitude can be improved when training a regularized model using CoNNect, rather than training it via standard L1 regularization. On a similar note, one could study if integrating CoNNect in established pruning approaches, such as Lottery Ticket Hypothesis-based pruning, can lead to better results. To better emphasize the positioning of our paper in terms of its contribution, we have revised the manuscript accordingly.
>
> **[RW3]:** We have included an example of CoNNect on GNNs in Appendix D.2 in the revised manuscript. Moreover, we have transformers covered through our implementation in LLM-pruner in Section 4.3. Also, we have implemented modern CV models, please see response [RW4] of reviewer SZg4.
>
> Due to limited time, we have not implemented CoNNect in RNNs, but one would have to account for the following. Observe that $W$ becomes a cyclic graph (through the recurrent connections), with infinitely long paths. It follows that $$\varphi(W) = \sum_{k=1}^\infty (\theta(W))^k = (I - \theta(W))^{-1} - I,$$ and we must show that this exists. Note that it is sufficient to ensure that the largest eigenvalue of $\theta(W)$ is strictly less than 1, which follows by assuming that the output layer does not have recurrent connections, as is common for RNNs.
>
> **[RW4]:** We agree that hyperparameters can have effects on regularizer behavior. Please note we already included several ablation results for the experiment in Section 4.1, which can now be found in Appendix D.1. Moreover, we have added several more by changing the regularizer coefficients. CoNNect appears to be very robust in terms of its hyperparameter and consistently outperforms L1 and L2 regularization. In fact, the weakest performance of CoNNect, see Figure 9(b), still manages to beat almost all cases of L1 and No regularization.

---

> ### Author Response · Authors · 2024-11-23
> **Response to Reviewer DVNq (Part 2)**
>
> **[RQ1]:** We appreciate the references provided by the reviewer and are willing to elaborate on their relationship in our paper. But the comparison with these two methods may not be entirely "apple-to-apple." SparseGPT is a complex iterative algorithm that always requires parameter updates during pruning. And Wanda focuses on unstructured pruning, where the pruned model may encounter issues such as discontinuous memory access during inference or require specialized hardware to address it. In contrast, in our experiments in Section 4.3, we only extend the loss function in the LLM-pruner to include CoNNect. Only such a small change already results in improved results, which argues for connectivity-based pruning. This is what makes CoNNect a conceptual contribution since it can be applied in pre-established pruning frameworks such as LLM-pruner. In fact, we believe integrating CoNNect in SparseGPT and Wanda can be an interesting direction for future work.
>
> We further provide results under more aggressive sparsities for the reviewer’s reference, where our method generally demonstrates an advantage. Experiments were set to prune 60% and 90% of the structures, ultimately resulting in sparsities of 47% and 70%, respectively. To align with SparsGPT/Wanda, we present the following accuracy results without normalization.
>
> **Table.1 Llama-7B Accuracy Results.**
> |   Pruning Ratio | Method    | boolq     | piqa      | hellaswag   | winogrande   | arc_easy   | arc_challenge   | openbookqa   |
> |-----------:|:----------|:----------|:----------|:------------|:-------------|:-----------|:----------------|:-------------|
> |        0.47| LLMPruner | **61.28** | 69.37     | 39.46       | 53.75        | 47.05      | 26.11           | 21.80        |
> |        0.47| CoNNect   | 60.58     | **69.64** | **40.63**   | **55.17**    | **50.97**  | **26.96**       | **22.40**    |
> |        0.70 | LLMPruner | **62.17** | 58.87     | 28.40       | **51.38**    | 36.11  | 20.90       | 15.20    |
> |        0.70 | CoNNect   | 55.05     | **59.79** | **28.53**   | 50.20        | **36.53**      | **21.50**           | **16.00**        |
>
> **[RQ2]:** We thank the reviewer for these suggestions. We test our method on other series and scales of models, and the results show that our method still demonstrates positive effects. More results on CV models can be found in our response [RW4] to reviewer SZg4.
>
> **Table.2 Vicuna-7B Accuracy Results**
> |   Pruning Ratio | Method    | boolq     | piqa      | hellaswag   | winogrande   | arc_easy   | arc_challenge   | openbookqa   |
> |-----------:|:----------|:----------|:----------|:------------|:-------------|:-----------|:----------------|:-------------|
> |        0   |      -     | 75.69     | 77.15     | 55.98       | 67.80        | 69.02      | 40.78           | 31.40        |
> |        0.40| LLMPruner | **61.90** | **71.87** | 43.00       | 56.20        | 54.21      | 29.35           | 24.20        |
> |        0.40| CoNNect   | 47.61     | 71.33     | **43.58**   | **57.14**    | **54.97**  | **29.69**       | **26.60**    |
>
> **Table.3 Llama-13B Accuracy Results**
> |   Pruning Ratio | Method    | boolq     | piqa      | hellaswag   | winogrande   | arc_easy   | arc_challenge   | openbookqa   |
> |-----------:|:----------|:----------|:----------|:------------|:-------------|:-----------|:----------------|:-------------|
> |        0   |           | 68.53     | 78.78     | 59.09       | 70.09        | 74.58      | 43.94           | 30.60        | 60.80     |
> |        0.42 | LLMPruner | 62.29     | 74.97     | 49.45       | 60.77        | 64.14      | 33.53           | 26.00        |
> |        0.42 | CoNNect   | **63.00** | **75.73** | **50.50**   | **61.40**    | **65.03**  | **33.79**       | **27.60**    |
>
>
> > References: Frankle, Jonathan, and Michael Carbin. "The lottery ticket hypothesis: Finding sparse, trainable neural networks." arXiv preprint arXiv:1803.03635 (2018).

---

> > ### Comment · Reviewer_DVNq · 2024-11-26
> >
> > I thank the authors for their reply to my previous questions and comments. I greatly appreciate the effort put into addressing them. However, I still have the following concerns:
> >
> > 1.  **Reference on use of Katz centrality for pruning**: The use of Katz centrality for pruning may not be entirely novel. For instance, see [1]. This reference appears to be missing, and a brief discussion of it would be helpful, as the article directly employs a similar idea (albeit more for neuron merging and other networks, but the concept remains similar).
> >
> > 2. **Performance Evaluation**:
> >    While I understand that the goal of CoNNect is to propose a plug-and-play regularizer for pruning neural networks, the key metric to judge objectively its impact is still the evaluation scores. At present, CoNNect combined with LLMPruner seems to fall short of competing with state-of-the-art pruning methods like **Wanda** and **SparseGPT** at similar levels of sparsity. Indeed to justify the conceptual framework of CoNNect it will be valuable to explore whether CoNNect could be integrated with these techniques to enhance its performance. But as the current evaluation scores seems to be not comparable to state of the art pruning techniques, I keep my score.
> >
> > ### Reference
> > [1] A pruning feedforward small-world neural network based on Katz centrality for nonlinear system modeling by Wenjing Li, Minghui Chu, Junfei Qiao in Neural Networks, Volume 130, October 2020, Pages 269-285.

---

> > > ### Author Response · Authors · 2024-11-29
> > > **Response to Reviewer DVNq**
> > >
> > > We sincerely thank the reviewer for their response. We hope to address the remaining two concerns below in a manner that allows for reassessment and a favorable outcome for our work.
> > >
> > > [R1] We agree that [1] deserves a discussion, as it explicitly refers to Katz centrality in network pruning. However, we find our work to be quite different when compared to [1], because it does not capture NN connecitivity, like we do. We provide the differences and improvements introduced by our work below.
> > >
> > > 1. [1] merges nodes based on Katz centrality, whereas our paper focuses solely on connectivity measurements derived from Katz centrality. The key distinction here lies in how node importance in feedforward NNs is computed: [1] evaluates importance based on the path weights **originating** from a node, whereas the CoNNect approach evaluates importance based on the path weights passing **through** a node. Both are very different measurements and this distinction is crucial, as, for example, the method in [1] cannot guarantee the prevention of layer collapse. Nodes with low overall weights may still be pruned in favor of Katz-central nodes that exhibit high weights on only one side.
> > > 2. We further modify the connectivity measurement by considering normalized weights and taking the absolute value of weights. Given that a NN can have both negative and positive weights, we are quite unsure how the approach in [1] can be effective in neural network pruning without considering the absolute value of weights.
> > >
> > > [R2] We would like to further clarify the distinctions between our approach in LLM-Pruner and methods such as Wanda and SparseGPT. SparseGPT and Wanda exclusively support unstructured or $n:m$ semi-structured pruning, while our integration in LLM-Pruner employs structural pruning. A key advantage of structural pruning is that it does not rely on specialized hardware (e.g., GPUs with the Ampere architecture) to achieve inference speed improvements. These settings can result in different tradeoffs between performance drop and inference speed-up.
> > >
> > > To be specific, while Wanda's $2:4$ semi-structured pruning achieves an approximate $1.6\times$ wall time speed-up, our method and LLMPruner delivers higher speed-ups of around $1.9\times$ at the same 50% sparsity level. Furthermore, our approach allows for structural pruning at higher sparsity ratios (as detailed in our earlier response), achieving near-optimal inference speed-ups for the given sparsity. In contrast, Wanda is limited to unstructured pruning in such scenarios, leading to comparatively lower improvements in inference speed.
> > >
> > > This distinction is critical because inference speed and accuracy performance represent the fundamental trade-off in neural network pruning, and both should be reflected in a model evaluation. While Wanda demonstrates good accuracy, its inference speed improvements are inferior to those achieved by CoNNect + LLM-Pruner. Moreover, Wanda's applicability is restricted to very specific settings (pruning ratios and specialized hardware), further highlighting the versatility and practical advantages of our approach. Therefore, when considering both accuracy and inference speed, CoNNect integrated with LLM-Pruner offers a competitive method among these state-of-the-art pruning methods.

---

> > > > ### Author Response · Authors · 2024-12-02
> > > > **Response to Reviewer DVNq**
> > > >
> > > > Dear Reviewer DVNq,
> > > >
> > > > Thank you for taking the time to review our paper. We truly value the feedback you have provided.
> > > >
> > > > With the discussion phase concluding soon, we kindly seek your input on our replies to your comments. If you need further clarification or have additional questions, please don’t hesitate to let us know.
> > > >
> > > > Best, Authors

---

> > > > ### Comment · Reviewer_DVNq · 2024-12-02
> > > >
> > > > I thank the authors for the detailed comments.
> > > >
> > > > 1. I believe the discussion pointed out in the rebuttal should be in the article in Related Works section and also the mentioned paper should have been (at least) cited. I understood the difference between the two articles but the underlying core theoretical idea is similar. As a reviewer my concern was more the missing citation.
> > > >
> > > > 2. The performance drop of CoNNect compared to the state of the art methods mentioned above is still substantial for the same level of sparsity. If the performance was similar with the slight gain in inference speed as mentioned by the authors, then yes. But as it stands, the slight gain in inference speed doesn't balance out the substantial drop in performance from the state of the art.

---

> > > > > ### Author Response · Authors · 2024-12-04
> > > > > **Response to Reviewer DVNq**
> > > > >
> > > > > Regarding the missing citation, we agree that the discussed paper can be included in the Related Works section. In a revised version of our manuscript, we will ensure that this paper is appropriately cited.
> > > > >
> > > > > On the matter of performance, as mentioned in our response, CoNNect targets a different aspect of pruning by also optimizing inference speed and hardware efficiency, rather than solely focusing on accuracy. While Wanda indeed exhibits a difference in accuracy compared to CoNNect, it is important to note that their baseline unpruned models also perform significantly better in terms of accuracy. This suggests that Wanda’s accuracy results might partially stem from the stronger starting point of their unpruned models, rather than the pruning method itself. Moreover, the gains in inference speed and hardware efficiency achieved by CoNNect can provide a valuable trade-off. **In fact, upon revisiting the Wanda paper, we found that only the linear layers of LLaMA-7B achieve a $1.6\times$ inference speed-up. However, the end-to-end latency speed-up for LLaMA-7B using Wanda is (only) around $1.2\times$ at 50% sparsity. Thus, their inference time is reduced by <20%. In contrast, CoNNect achieves a $1.9\times$ speed-up, nearly halving the inference time and approaching the theoretical maximum of $2\times$ for 50% sparsity.**
> > > > >
> > > > > Furthermore, our contribution is broader than only pruning LLMs, as CoNNect shows good results across a diverse range of neural networks, e.g., MLP, CNN (ResNet/VGG), GNN, Transformer (LLM), which we hope will spark further exploration of CoNNect pruning in this area.
> > > > >
> > > > > Sincerely,
> > > > > Authors

---

### Official Review · Reviewer_eRZy · 2024-10-27

**Soundness:** 3
**Presentation:** 3
**Contribution:** 3
**Rating:** 6
**Confidence:** 4

**Summary:**

This work proposed CoNNect, a regularization-based neuron connectiviy-preseving neural network pruning algorithm. Specifically, the authors consider the computation graph of neural network as a directed weighted graph, where the each node represents a neuron, each weight parameter represents a weighted-edge. The CoNNect regularizer penalizes the the negative logarithm of the connectivity between the input layer and the output layer. The authors proved that 1) Theorem 1: CoNNect does incur sparsity, 2) Theorem 2: the stationary points of CoNNect regularizer are its global minimizers. The authors validate the effectiveness of CoNNect on MLP on synthesis regression data, VGG-11 on CIFAR-10, and LLaMA-7B on multiple datasets.

**Strengths:**

1. The authors introduce a principled algorithm (CoNNect regularizer) to preserve the output-input layer connectivity during pruning.
2. The authors provide theoretical guarantee on the sparsity of CoNNect minizer, and characterize its stationary points.
3. The authors validate the effectiveness of CoNNect on MLP, VGG, and LLaMA-transformer.

**Weaknesses:**

1. The connectivity analysis did not explicitly consider the residual connection structure, which is ubiquitous in mainstream machine learning models, e.g. ResNet, UNet, and Transformers. When the residual connection is involved, the input and output features are ensured to be connected. In this case, it seems that existing pruning methods can still enjoy a satisfactory connectivity between the input and output layer. I wonder to what extend does the residual connection structure overshadow the necessity and benefit of the proposed CoNNect regularizer (which may hinder the training process).

2. Beside the standard gradient-based $L_1$ regularization, the authors are recommended to compare their method with improved $L_1$ optimization methods, e.g. Lasso shrinkage operator-based optimization [1] and Spred-$L_1$ [2].

3. According to Figure 4, it seems that the CoNNect regularizer hinders the train from scratch process of the model, as the `No Reg. w/ Tun` consistently outperforms the `CoNNect Reg. w/ Tun.` until a pruning ration below $0.8$.

4. Minor Typo:

   a) In Page 4, lines 207-210, should $V_k$ be captalized as $V_K$?

   b) Page 6, lines 311-312, magnetude-based.



[1]. Tibshirani, R. (1996). Regression shrinkage and selection via the lasso. Journal of the Royal Statistical Society: Series B (Methodological), 58(1):267–288.

[2]. Liu Ziyin and Zihao Wang. 2023. Spred: solving L1 penalty with SGD. (ICML'23), Vol. 202.

**Questions:**

1. Can the authors provide the detailed implementation of the calculation of the total conectivity regularizer Eq. (3) for mainstream architectures (e.g. VGG, ResNet, Transformer, UNet)? Do we need to parse the network into an adjacency matrix representation, or there is a more efficient and practicable implementaion?

2. Can the authors compare the FLOPs and inference acceleration of the pruned models against the baseline methods? Can the authors provide a complexity analysis of the proposed algorithm?

---

> ### Author Response · Authors · 2024-11-23
> **Response to Reviewer eRZy**
>
> **[RW1]:** We thank the reviewer (and reviewer SZg4) for this comment on residual neural networks. We mainly omitted the residual connection in the analysis because it makes the presentation of the theoretical results easier.
>
> Simply connecting input and output can only ensure that the network avoids layer collapse, but it does not further aid in model compression. While residual connections can ensure a direct link between input and output features, CoNNect can be used to further enhance the connectivity rather than replace these connections. Despite the existence of residual connections, the backbone's processing of input information remains indispensable; otherwise, the entire network would degrade. CoNNect enhances the overall connectivity of the network by optimizing the parameters of the backbone, thereby facilitating the compression of the backbone modules without hindering the performance too much.
>
> We have included a detailed implementation of CoNNect in Appendix B, also highlighting the case of residual connections. Regularizing with CoNNect is feasible because residual connections are typically non-parameterized and therefore won't bother the optimization process when enhancing the network's connectivity.
>
> To further demonstrate CoNNect on residual neural networks, we apply CoNNect on ResNet-56, please see our response [RW4] to reviewer SZg4 for these results.
>
> **[RW2]:** We thank the reviewer for pointing out that we should compare CoNNect with the newest L1 regularization methods and have added these references in our revision. We believe we have in fact implemented the loss of shrinkage operator in our current numerical results. To study the competitiveness of CoNNect against Spred-L1 regularization, we have applied Spred-L1 in the numerical example in Section 4.1 and see that its performance indeed improves upon standard L1 regularization and matches CoNNect's performance. However, this comes at the cost of doubling the number of trainable weights, which increases the time and space complexity, whereas CoNNect does not suffer from such an increase in complexity. We also have compared Spred-L1 in the same manner of Section 4.2 by following the official implementation and still highlight our superiority over it in the table below.
>
> **Table.1 Results of CoNNect and Spred-L1 for training full-scale VGG-11 on CIFAR-10.**
> |Method|Pruning Ratio | 0.5 | 0.6 | 0.7 | 0.8 | 0.9|
> |:-:|:-:|:-:|:-:|:-:|:-:|:-:|
> |CoNNect|w/o finetuing|74.09|74.09|74.09|74.09|45.62|
> | |w/ finetuing|83.20|83.25|83.30|83.25|81.33|
> |Spred-L1|w/o finetuing|38.85|20.28|17.06|10.00|10.00|
> | |w/ finetuing|81.03|80.45|79.61|74.42|74.65|
>
> **[RW3]:** We chose the CoNNect hyperparameter $\lambda_2$, see (7) in the revision, such that we can show we obtain good results at high pruning ratios. To show better results in lower pruning ratios, one can simply decrease the value of our $\lambda_2$ (where for $\lambda_2 \to 0$, we approach the performance non-regularized model).
>
> **[RW4]:** Thank you for pointing out both typos. We improved our manuscript accordingly.
>
> **[RQ1]:** In the manuscript, we present CoNNect via the adjacency matrix because that is how the connectivity measurement in Katz centrality is initially defined. However, in our implementation of CoNNect we do not use the adjacency matrix. Instead, it suffices to define a slightly adapted forward function of the neural network. We  (for more implementation details, see Appendix B in the revised manuscript). The time complexity of the adjusted forward function is bounded by the original forward function, see also our response [RQ1] for reviewer SZg4. That means that for a batch size of $M$, the time used for computing CoNNect is proportional to $1/M$, which is for modern batch sizes a very moderate computational cost.
>
> **[RQ2]:** We evaluate the computational overhead of LLM from Section 4.3 in terms of MACs. Since we share a similar grouping strategy, we can achieve inference acceleration comparable to LLMPruner. More results on CV models can be found in our response [RW4] to reviewer SZg4.
>
> **Table.2 Statistics of the base model and the compressed model.**
> | Pruning Ratio | # Para | GPU Memory | Computational complexity| Speed up |
> |:--:|:--:|:--:|:--:|:--:|
> |0   |6.7B| 12892.64MiB| 425.1GMACs|-|
> |0.40|4.1B| 7952.64MiB| 255.82GMACs|1.66$\times$|
> |0.47|3.6B| 6995.72MiB| 222.67GMACs|1.91$\times$|
> |0.70|2.0B| 3942.96MiB| 123.23GMACs|3.45$\times$|

---

> ### Comment · Reviewer_eRZy · 2024-11-24
>
> Dear authors,
>
> Thanks for your time and effort in providing detailed response and extra experiments. As my W2, W3, W4, Q1, and Q2 are addressed  I raise my score from (rating = 5, conf = 3) to (rating = 6, conf = 4).
>
> I am not considering a higher score, because the current analysis does not apply to residual structures (which is ubiquitous in modern deep neural networks). This limitation weakens the contribution of this paper.
>
> Best,
>
> Reviewer eRzy

---

> > ### Author Response · Authors · 2024-11-25
> > **Response to Reviewer eRZy**
> >
> > Dear Reviewer eRzy,
> >
> > Thank you for your reconsideration and for raising your score.
> >
> > We would like to kindly clarify that the current analysis does apply to residual structures, as detailed in Appendix B of the revised manuscript. Furthermore, we have demonstrated the application of CoNNect on residual neural networks, specifically using ResNet-56, in our response to Reviewer SZg4 ([RW4] in the discussion).
> >
> > We hope this addresses your concern regarding the general applicability of our analysis. Please let us know if you have further questions or require additional clarification.
> >
> > Best, Authors

---

> ### Comment · Reviewer_eRZy · 2024-11-25
>
> Dear authors,
>
> I would like to clarify my previous comment: my initially opinion is that, the current 'connectivity modeling' did not explicitly take the skip connection into account, and it just treat the residual shortcut as a constant and omit its effect on connectivity.
>
> From my understanding, the residual shortcut is indeed an unprunable weight that significantly affects the global connectivity. Let's consider a residual MLP given by the code attached below, if we prune all the neurons in `self.hidden_layer1`, the connectivity edge between neurons from `self.input_layer` and `self.hidden_layer2` is still nonzero, as they are connected by the shortcut. This fundamentally change the adjacency matrix induced by the neurons / weights. I guess this can also change the property of the stationary points of the penalty.
>
>
> Code attachment:
>
> ```{python}
> import torch
> import torch.nn as nn
> import torch.nn.functional as F
>
> class ResidualMLP(nn.Module):
>     def __init__(self, input_dim, hidden_dim, output_dim):
>         super(ResidualMLP, self).__init__()
>         self.input_layer = nn.Linear(input_dim, hidden_dim)
>         self.hidden_layer1 = nn.Linear(hidden_dim, hidden_dim)
>         self.hidden_layer2 = nn.Linear(hidden_dim, hidden_dim)
>         self.output_layer = nn.Linear(hidden_dim, output_dim)
>         self.activation = nn.ReLU()
>
>     def forward(self, x):
>         # Input layer
>         x = self.activation(self.input_layer(x))
>
>         # Residual block 1
>         residual = x
>         x = self.activation(self.hidden_layer1(x))
>         x += residual  # Add skip connection
>
>         # Residual block 2
>         residual = x
>         x = self.activation(self.hidden_layer2(x))
>         x += residual  # Add skip connection
>
>         # Output layer
>         x = self.output_layer(x)
>         return x
> ```

---

> > ### Author Response · Authors · 2024-11-25
> > **Response to Reviewer eRZy**
> >
> > Thank you for providing further clarification on your comment. We greatly appreciate your effort and now better understand your perspective.
> >
> > It seems that we differ in our design choice for constructing the "connectivity graph" used to compute a neural network’s connectivity $\varphi^{tot}(W)$. If we understand correctly, you propose that connectivity should propagate through skip connections and therefore be included in this connectivity graph. Following your example, one would then compute the CoNNect regularizer:
> >
> > ```
> > def CoNNect_forward(self, x): # input x is vector of ones.
> >
> >     # Normalize weights via Equation (2)
> >     self.normalize_weights()
> >
> >     # Input layer
> >     x = self.input_layer(x)
> >
> >     # Residual block 1
> >     residual = x
> >     x = self.hidden_layer1(x)
> >     x += residual  # Add skip connection
> >
> >     # Residual block 2
> >     residual = x
> >     x = self.hidden_layer2(x)
> >     x += residual  # Add skip connection
> >
> >     # Output layer
> >     x = self.output_layer(x)
> >
> >     # Recover original weights
> >     self.denormalize_weights()
> >
> >     return torch.sum(x)
> > ```
> >
> > And our theoretical analysis exclusively utilizes learnable weights, thereby omitting skip connections in the "connectivity graph". This can be calculated as:
> >
> > ```
> > def CoNNect_forward(self, x): # input x is vector of ones.
> >
> >     # Normalize weights via Equation (2)
> >     self.normalize_weights()
> >
> >     # Input layer
> >     x = self.input_layer(x)
> >
> >     # Residual block 1
> >     x = self.hidden_layer1(x)
> >
> >     # Residual block 2
> >     x = self.hidden_layer2(x)
> >
> >     # Output layer
> >     x = self.output_layer(x)
> >
> >     # Recover original weights
> >     self.denormalize_weights()
> >
> >     return torch.sum(x)
> > ```
> >
> > We view this as a design decision that parallels other choices, such as whether to include activation functions or biases in connectivity computations. As such, we see this flexibility as a feature of CoNNect, allowing future research to iterate and refine its operation for neural network pruning. In a revision, we will ensure this flexibility is communicated more clearly.
> >
> > That said, we believe both approaches are valid for pruning residual neural networks, and both design choices won't imply an inability to prune such networks. To further illustrate this point, we applied residual connections from hidden layer 1 to hidden layer 2 and from hidden layer 2 to hidden layer 3 in the numerical example provided in Section 4.1. Notably, both approaches performed well.
> >
> > Ultimately, we believe further empirical validation is needed to determine the optimal configuration of CoNNect for neural network pruning, including whether to incorporate residual connections, biases, or activation functions in the connectivity graph.
> >
> > We sincerely thank you for this observation, which has contributed to improving the quality of this manuscript. We hope this explanation provides clarity and addresses your concern. Please don’t hesitate to reach out with any additional questions or comments.

---

> > > ### Author Response · Authors · 2024-12-02
> > >
> > > Dear Reviewer eRZy,
> > >
> > > Thank you for your review of our paper. We truly appreciate the time and effort in helping to improve our manuscript.
> > >
> > > With just one day remaining in the discussion phase, we kindly request your feedback on our latest response. Should you have any further questions or need clarification, please don’t hesitate to reach out. We're happy to provide any additional information.
> > >
> > > Best, Authors

---

### Official Review · Reviewer_oiEe · 2024-11-04

**Soundness:** 3
**Presentation:** 3
**Contribution:** 3
**Rating:** 6
**Confidence:** 1

**Summary:**

The paper introduces a new neural network pruner CoNNect that focuses on maintaining connectivity between network layers. CoNNect achieves this by leveraging Katz centrality to ensure sparse yet fully connected networks. The CoNNect regularizer is a differentiable and effective surrogate for the L0 norm, promoting sparse architectures while preserving crucial connections. CoNNect avoids issues like layer collapse by maintaining a stable structure, beneficial for both unstructured and structured pruning. Experiments demonstrate that Connect performs well on the LLMs.

**Strengths:**

1. The paper is overall well-written and logically well-organized.
2. The CoNNect is proven to approximate the L0-norm regularization, and the pruned neural network architecture guarantees the maximal connected network structure, which seems solid.
3. The CoNNect is differentiable and can be optimized by gradient descent.

**Weaknesses:**

1. In the introduction, the author highlights that neural network connectivity is important for good running. However, the benefits of maintaining such connectivity for pruning are unclear to readers. It would be better if the authors could illustrate this further.

2. In Section 3.3.1, the motivation to use Katz centrality for neural network pruning can be illustrated better. Specifically, while there are many centrality measures in network analysis, it is unclear why Katz centrality stands out.

3. In Section 4.1, is it necessary to prune a small MLP model for evaluation?

4. Some typos, e.g. caries -> carries in Line 348.

**Questions:**

Please refer to weaknesses. Because the reviewer is not an expert, the novelty and the correctness of the proofs cannot be evaluated.

---

> ### Author Response · Authors · 2024-11-23
> **Response to Reviewer oiEe**
>
> **[RW1]:** We agree with the reviewer that Axiom 2 can use improved motivation. During pruning, it can happen that many or all connections between two layers are pruned, rendering the pruned model to be useless. We have illustrated this using a figure in the introduction. We also slightly revised the text to emphasize this point. Moreover, through a series of numerical experiments, we demonstrate how incorporating CoNNect into existing pruning methods enhances neural network connectivity and yields good results.
>
> **[RW2]:** Please note it isn't Katz centrality but the connectivity measurement in Katz centrality that we are using. This form of connectivity measurement allows us to favor a few direct paths over many parallel paths. We do not see how other connectivity metrics in graph theory can achieve the same result.
>
> **[RW3]:** The goal of the small MLP model in section 4.1 is to provide a small example that illustrates the working of CoNNect. We believe that it nicely illustrates how typical regularization approaches fail in finding the sparsest network representation, whereas CoNNect much more often succeeds. Moreover, to strengthen our results, we performed many ablation tests in Appendix D.1 and see that CoNNect is much stronger than L1 and No regularization.
>
> **[RW4]:** We thank the reviewer for pointing out this typo. We have adjusted the manuscript accordingly.

---

> > ### Comment · Reviewer_oiEe · 2024-11-24
> > **Thanks for the rebuttal**
> >
> > Thank the authors for the rebuttal. For RW3, the motivation may not be convincing. The technical sections have introduced how CoNNect works. But why is a small MLP still required to illustrate this?

---

> > > ### Author Response · Authors · 2024-11-25
> > > **Response to Reviewer oiEe**
> > >
> > > Thank you for your response. The small MLP example in Section 4.1 is meant to provide intuition and a clear visual illustration of the effects of the CoNNect regularizer. A larger network would not serve this purpose as well, as the results would be harder to interpret and visualize. We believe this example supports the more detailed quantitative analysis in later sections by offering a demonstration of CoNNect’s strengths.

---

> > > > ### Author Response · Authors · 2024-12-02
> > > > **Response to Reviewer oiEe**
> > > >
> > > > Dear Reviewer oiEe,
> > > >
> > > > Thank you for your review of our paper.
> > > >
> > > > With just one day remaining in the discussion phase, we would like to hear if any questions or needs for clarification are remaining. In that case, please don’t hesitate to reach out. We're happy to provide any additional information.
> > > >
> > > > Best, Authors

---

### Official Review · Reviewer_SZg4 · 2024-11-04

**Soundness:** 2
**Presentation:** 2
**Contribution:** 3
**Rating:** 6
**Confidence:** 3

**Summary:**

This paper studies the problem of neural network pruning. The authors propose two axioms for pruning: reducing the NNs' size by removing unnecessary weights and preserving their connectivity to ensure stable information flow. To address these two axioms, the authors propose a novel regularizer called CoNNect, which computes a connectivity matrix based on Katz centrality, providing a measure that prefers direct paths over parallel ones. The method demonstrates higher performance in various pruning scenarios, including VGG-11 for CIFAR-10 and large language models (LLMs) like LLaMA-7B for various NLP tasks.

**Strengths:**

1. It is a novel use of Katz centrality to enhance pruning by maintaining connectivity, which is an overlooked factor in neural network pruning. This approach ensures that information can effectively flow from input to output layers, addressing the common issue of layer collapse in sparse networks.

2. The authors develop a strong theoretical foundation, proving that CoNNect approximates L0 regularization while preventing collapse, which strengthens the validity of their approach.

3. Through comprehensive experiments, the authors provide evidence of CoNNect’s superiority over standard L1- and L2-based regularization techniques. The approach's effectiveness in CV and NLP benchmarks demonstrates the practical impact of this research.

4. The writing is clear and well-structured, making it easy to follow the authors’ arguments and contributions.

**Weaknesses:**

1. The authors should discuss more about the second axiom (Preserve Neural Network Connectivity). Why is preserving connectivity important for pruning? How could keeping the flow of information benefit the performance of the pruned model? More insights on this point would make the paper more convincing.

2. The convergence analysis for regularizer loss in lines 243-280 seems impractical and unnecessary, since the parameters not only receive gradients from the regularizer but also from the main task loss. Then these two gradients will added together. With this different gradient, will it still converge to minimal point of regularizer loss?

3. This work does not apply to some important architectures like resnet with residual connections. The authors could provide some discussions on possible solutions for these architectures.

4. The experiments are not extensive. If the authors could provide more results on some modern CV models and datasets, it would be more convincing.

**Questions:**

My main concerns and suggestions are already mentioned in the weaknesses section. I would like to mention some minor points here.

1. Time/space complexity analysis could be provided.

2. The presentation could be improved. I think Figure 2 could serve as a perfect illustration of intuition, thus I suggest drawing a similar one in the introduction part.

---

> ### Author Response · Authors · 2024-11-23
> **Response to Reviewer SZg4**
>
> **[RW1]:** We thank the reviewer for this comment. Neural network connectivity is crucial, as without it, the input signal cannot reach the output, typically resulting in a non-functional neural network. We have illustrated this by slightly revising the introduction and inserting Figure 1 after Axiom 2 in the revised manuscript. Moreover, we show through various numerical experiments how integrating CoNNect in existing pruning methods improves neural network connectivity. The good results emphasize the role of Axiom 2 in ensuring the network’s functionality.
>
> **[RW2]:** The goal of this analysis is to show that all stationary points induced by the CoNNect regularizer are unstable, except for the global minimizers. What our result shows is that CoNNect itself does not contain any stable stationary points that can serve as an attraction region. In case the algorithm gets stuck in a stationary point of the regularizer, the loss function will always push the solution to leave the stationary point unless the loss function itself is stationary at that point.
>
> **[RW3]:** Applying CoNNect to residual neural networks is in fact quite straightforward. The identity connection itself does not contain any trainable parameters and thus can be ignored when regularizing with CoNNect. Based on the reviewer's suggestion, we now provide a detailed implementation of CoNNect in Appendix B. Moreover, please see our response at [RW4] for an application on modern CV models.
>
> **[RW4]:** Similarly to Section 4.3, we integrate CoNNect into the preceding work of LLMPruner, DepGraph (Fang et al., 2023), and obtain the following results. DepGraph is also a structured pruning method that can be based on Taylor's importance. We iteratively prune until the predefined speed-up targets, such as 2.5$\times$, 8$\times$, or 16$\times$, are achieved, which is calculated as the ratio of MACs before and after pruning.
>
> **Table.1 Results on CV models.**
> | Setting | Base Acc.| Method | Pruned acc. | Speed Up | Pruning Ratio |
> |:-----------:|:----------:|:------------:|:----------:|:------------:|:-------------:|
> |ResNet56-CIFAR10|93.53| DepGraph | 93.17 | 2.51$\times$ | 0.56 |
> ||| CoNNect | **93.63** | 2.50$\times$ | 0.53 |
> ||| DepGraph | 80.24 | 16.17$\times$ | 0.98 |
> ||| CoNNect | **83.12** |  17.24$\times$ | 0.97 |
> |VGG19-CIFAR100|73.50| DepGraph | 65.89 | 8.12$\times$ | 0.90 |
> ||| CoNNect | **69.38** | 8.00$\times$ | 0.93 |
> ||| DepGraph | 57.48  |  16.10$\times$ | 0.96 |
> ||| CoNNect | **62.56** |  16.07$\times$ | 0.98 |
>
> **[RQ1]:** Fortunately, the time complexity of CoNNect is bounded by a single forward pass of the NN. Thus, for a batch size of $M$, the additional time used for computing the CoNNect regularizer is proportional to $1/M$. Moreover, the space complexity is negligible since we only have to define a slightly adapted forward pass function for the same neural network and no storage of additional parameters is required. We have emphasized this in the revised manuscripts on lines 310-314.
>
> **[RQ2]:** We thank the reviewer for finding our illustration helpful. We have added a figure showcasing layer collapse in the introduction.
>
> > References: Fang, Gongfan, Xinyin Ma, Mingli Song, Michael Bi Mi, and Xinchao Wang  "Depgraph: Towards any structural pruning." Proceedings of the IEEE/CVF Conference on Computer Vision and Pattern Recognition (2023): 16091-16101

---

> > ### Author Response · Authors · 2024-11-26
> > **Response to Reviewer SZg4**
> >
> > Dear Reviewer SZg4,
> >
> > We have submitted our response to your questions and revised our paper in line with your suggestions. We sincerely appreciate your valuable feedback, which has helped us improve the quality of our work.
> >
> > We hope the revisions address your concerns and meet your expectations. Please let us know if there are any additional questions or aspects you'd like us to address.
> >
> > Best, Authors

---

> > > ### Comment · Reviewer_SZg4 · 2024-11-27
> > >
> > > Thanks for your rebuttal. My concerns are addressed, therefore I decide to keep my positive score.
> > >
> > > Regards,
> > > Reviewer SZg4.

---

> > > > ### Author Response · Authors · 2024-12-02
> > > > **Response to Reviewer SZg4**
> > > >
> > > > Dear Reviewer SZg4,
> > > >
> > > > Thank you for the time invested in reviewing our manuscript. If there are any concerns remaining, please feel free to let us know.
> > > >
> > > > Best,
> > > > Authors

---

### Official Review · Reviewer_JAt6 · 2024-11-07

**Soundness:** 2
**Presentation:** 2
**Contribution:** 1
**Rating:** 3
**Confidence:** 5

**Summary:**

The focus of the paper is to develop a pruning strategy for a sparse network. This has been
achieved by the differentiable regularizer. The approach maintains connectivity and prevents
layer collapse. This is an important and topical area of research. The results from the approach
in the paper, named CoNNect has been compared with L1 norm regularizer and used for one-shot
pruning in LLMs.

**Strengths:**

1. A proof that the approach approximated the L0 regularizer  and guarantees maximally connected
structures as a stationary points.

**Weaknesses:**

1. Several research work on pruning has aimed at and succeeded in accomplishing
even more successfully what has been achieved here.
2. A comparison with the existing body of literature is needed as maximal connectivity,
avoiding layer collapse and zero shot pruning have also been addressed previously.
So the contribution of this work in the correct context is not emphasized.
3. The CoNNect is only compared with magnitude pruning and Synflow. Other pruning
approaches have not been considered for comparison.
4. The experimental results in Table 2 doesn't show any significant difference
with CoNNect as compared to others. The performance also falls significantly
with pruning.

**Questions:**

1. How are the theoretical results reflected in the experimental results?
A more through analysis perhaps can give more insight about the impact of
the regularizer.

---

> ### Author Response · Authors · 2024-11-23
> **Response to Reviewer JAt6**
>
> **[RW1]:** We appreciate the reviewer's comment and would benefit from specific guidance on this point. Are there any specific works that you would like us to discuss in the manuscript?
>
> **[RW2]:** We understand the reviewer's concern. However, even after an extensive literature search, we do not see to what references the reviewer refers to.
>
> In terms of the paper's contribution, we think that CoNNect provides a conceptual contribution to neural network pruning and can enhance existing pruning methods by improving neural network connectivity (Axiom 2). CoNNect's versatility is demonstrated via a soft pruning approach (e.g. regularization), and, as we show in Section 4.3, it is easily integrated into hard pruning approaches, such as one-shot pruning via LLM-pruner (Ma et al., 2023). Our positive numerical results highlight the importance of connectivity in neural network performance, stressing the relevance of Axiom 2.
>
> **[RW3]:** Following up upon our previous comment [RW2], we merely want to demonstrate how CoNNect can complement current neural network pruning approaches. Moreover, it is not that we are conducting a comparison between CoNNect and methods such as magnitude pruning. Instead, we show how a magnitude pruning can benefit from using CoNNect regularization as opposed to traditional methods such as L1 (see Section 4.1). We have emphasized this in the revision to avoid such confusion.
>
> In terms of integration of CoNNect within various pruning methods, we would like to highlight that CoNNect shows good results versus DepGraph (see our reply [RW4] on reviewer SZg4 for these results) and LLM-pruner, both of which have been compared with a large variety of methods and have outperformed these. This implies that CoNNect is also competitive against these methods.
>
> **[RW4]:** We believe the reviewer may have misread Table 2, as it does show significant differences. We improve the results of LLM-pruner (Ma et al., 2023), a highly cited paper in the field, in almost all cases via integrating CoNNect. Moreover, the differences are comparable with those achieved in the LLM-pruner paper. We have simplified the presentation in Table 2 in the revision and have stressed these results in the text. We hope any remaining confusion is resolved.
>
> **[RQ1]:** We can best highlight theoretical results in terms of the numerical example in Section 4.1. What we show is that CoNNect outperforms benchmark regularization methods when performing magnitude and SynFlow pruning. In this example, it becomes clear that traditional regularization methods suffer from layer collapse at high sparsity levels when pruning, or even already during training. Instead, CoNNect manages to train the NN in such a way that the weights can be pruned according to a simple magnitude-based pruning strategy. In other words, it is trained to prevent layer collapse and so it is guaranteed to satisfy Axiom 2.
>
> > References:  Ma, Xinyin, Gongfan Fang, and Xinchao Wang. "Llm-pruner: On the structural pruning of large language models." Advances in neural information processing systems 36 (2023): 21702-21720.

---

> ### Comment · Reviewer_eRZy · 2024-11-25
>
> Dear Reviewer JAt6,
>
> As I have also been working on connectivity enhancing pruning. I am also curious about, what are the latest works that perform the best in accomplishing connectivity enhancement in pruning? Would you please provide some reference?
>
> Best,
>
> Reviewer eRZy

---

> > ### Author Response · Authors · 2024-11-26
> > **Response to Reviewer JAt6**
> >
> > Dear Reviewer JAt6,
> >
> > We wanted to kindly follow up on our responses to your comments on the manuscript. Please let us know if you have any additional feedback or clarifications.
> >
> > Thank you for your time and valuable insights.
> >
> > Best, Authors

---

> ### Comment · Reviewer_JAt6 · 2024-11-26
> **Comment on the response**
>
> The reviewer would like to thank the authors for the clarifications.
>
> However, a fundamental issue still remains and that is the performance of the proposed approach when compared with
> other recently proposed approaches in pruning (such as expander based 'Pal et al. Revisiting the Lottery Ticket Hypothesis: A Ramanujan Graph Perspective ICLR 2022', 'Hoang et al. Revisiting Pruning at Initialization through the Lens of Ramanujan Graph ICLR 2023  or weights and activation based 'Sun et al. A Simple and Effective Pruning Approach for Large Language Models ICLR 2024').
>
> Even though the approach based on the proposed regularizer may be a conceptual contribution, the end results are not convincing yet. Also, it turns out that the concept of Katz centrality is not entirely novel as similar concept as in the paper has been
> available in the existing literature.

---

> ### Author Response · Authors · 2024-11-29
> **Response to Reviewer JAt6**
>
> We appreciate the reviewer's acknowledgment of our clarifications. We hope to address the remaining fundamental issue below.
>
> We have consulted the suggested papers and compared their performance with CoNNect results. Pal et al. (2022) mostly shows results on older architectures, such as Lenet. They provide a single result for VGG-19 on CIFAR-10 (see their appendix A.5). For comparison, we apply our CoNNect implementation via DepGraph (Fang et al., 2023) and show how we consistently outperform in Table 1. Please note that the final density is not exactly controllable and one must aim for a target MAC speed-up (the ratio of results before and after pruning). We refer to Fang et al. (2023) for further reference.
>
> Table.1 Results of VGG-19 on CIFAR-10 dataset by CoNNect.
> |MAC Speed-up|2.5$\times$|5$\times$|8$\times$|16$\times$|
> |:---:|:---:|:---:|:---:|:---:|
> |Density (=1-Sparsity)|22.52%|9.53%|6.12%|3.65%|
> |Accuracy|94.04%|93.55%|92.71%|91.22%|
>
> Comparing us to Hoang et al. (2023), we show how CoNNect outperforms all of their ResNet-34 implementations. Even while considering that the approaches used in Hoang et al. (2023) are unstructured pruning, which are known to perform better in accuracy than structured approaches, see Table 2.
>
> Table.2 Results of ResNet-34 on CIFAR-10 dataset by CoNNect.
> |MAC Speed-up|2.5$\times$|5$\times$|8$\times$|16$\times$|
> |:---:|:---:|:---:|:---:|:---:|
> |Density (=1-Sparsity)|30.13%|14.98%|9.16%|4.11%|
> |Accuracy|95.33%|94.55%|93.04%|91.56%|
>
> Considering the Wanda paper (Sun et al., 2023), we want to highlight that Wanda's performance is inferior to ours in terms of inference speed-up. Given that Wanda utilizes an $n:m$ semi-structural pruning, they achieve a wall time speed-up of around $1.6 \times$ at 50% sparsity level, where CoNNect + LLM-pruner performs a structural pruning, achieving a speed-up of around $1.9\times$ at the same 50% sparsity level. For more details, we would like to refer the reviewer to our response [R2] to Reviewer DVNq (at 29 Nov 2024, 12:09 CET).
>
> Finally, we would like to explain that CoNNect does **not** utilize Katz centrality for pruning, but merely the connectivity measurement employed in Katz centrality, which is a fundamental difference. Assuming the reviewer refers to Li et al. (2020), who use Katz centrality in NN pruning, we would like to refer the reviewer to our response [R1] to Reviewer DVNq (at 29 Nov 2024, 12:09 CET) to highlight our differences with this work.
>
> We hope these comments made our results more convincing and will lead to a favorable score. Please do not hesitate to reach out if further clarification is needed.
>
> > References:
> >
> > Fang, Gongfan, Xinyin Ma, Mingli Song, Michael Bi Mi, and Xinchao Wang  "Depgraph: Towards any structural pruning." Proceedings of the IEEE/CVF Conference on Computer Vision and Pattern Recognition (2023): 16091-16101
> >
> > Sun, Mingjie, Zhuang Liu, Anna Bair, and J. Zico Kolter. "A simple and effective pruning approach for large language models." arXiv preprint arXiv:2306.11695 (2023).
> >
> > Li, Wenjing, Minghui Chu, and Junfei Qiao. "A pruning feedforward small-world neural network based on Katz centrality for nonlinear system modeling." Neural Networks 130 (2020): 269-285.

---

> > ### Author Response · Authors · 2024-12-02
> > **Response to Reviewer JAt6**
> >
> > Dear Reviewer JAt6,
> >
> > Thank you for your review of our paper. We truly appreciate the time and effort you've dedicated.
> >
> > With just one day remaining in the discussion phase, we kindly request your feedback on our responses. Your insights are invaluable, and we look forward to your updated evaluation. Should you have any further questions or need clarification, please don’t hesitate to reach out. We're happy to provide any additional information.
> >
> > Best regards,
> >
> > Authors

---

> > > ### Comment · Reviewer_JAt6 · 2024-12-03
> > > **Comment**
> > >
> > > Thanks to authors for the clarifications and explanations. I have no further questions but still the results which
> > > indicate drop in performance accuracy without being offset by any significant benefit are not convincing. Hence,
> > > I decide against changing the score.

---

> > > > ### Author Response · Authors · 2024-12-04
> > > > **Response to Reviewer JAt6**
> > > >
> > > > While we acknowledge that Wanda (Sun et al., 2023) demonstrates higher accuracy, it is important to point out that their unpruned baseline models also exhibit significantly better performance in terms of accuracy. This suggests that a significant part of Wanda’s accuracy results could be attributed to their stronger starting point, rather than the pruning method itself. Moreover, we can balance this trade-off by better inference speed and hardware efficiency. **When re-evaluating the Wanda paper, we found that the $1.6\times$ inference speed-up applies only to the linear layers of LLaMA-7B and its the overall end-to-end latency speed-up is (only) approximately $1.2\times$ for 50% sparsity. Thus, their inference time is reduced by <20%. In comparison, CoNNect delivers a $1.9\times$ speed-up, reducing inference time by nearly half, which aligns with the 50% sparsity rate**. This highlights CoNNect's clear advantages in scenarios where computational efficiency is a priority.
> > > >
> > > > Moreover, we offer generality in our pruning approach, resulting in improved performances of traditional benchmarks on a wide variety of neural networks, e.g., MLP, CNN (ResNet/VGG), GNN, Transformer (LLM), which we believe has the potential to inspire further research in this field.
> > > >
> > > > We respect your decision to maintain your score but hope that our explanation provides additional context for the value of our contribution.
> > > >
> > > > Sincerely,
> > > > Authors
> > > >
> > > > > References:
> > > > >
> > > > > Sun, Mingjie, Zhuang Liu, Anna Bair, and J. Zico Kolter. "A simple and effective pruning approach for large language models." arXiv preprint arXiv:2306.11695 (2023).

---

### Author Response · Authors · 2024-11-23
**General Response**

We sincerely thank all reviewers for their thoughtful feedback and constructive comments. We are pleased that most reviewers recognized the originality of CoNNect and its strong theoretical foundation for neural network pruning. Some concerns were raised regarding its empirical application, such as the implementation complexity, and generalizability of our approach. Furthermore, additional experiments were suggested.

The following is a non-exhaustive list of revisions made in response to the reviewers. In particular, we have emphasized the generalizability of CoNNect to modern neural network architectures by including a detailed implementation section in Appendix B. Additionally, we conducted additional experiments on GNNs (see Appendix D.2), modern CV models (see Appendix D.3, or response [RW4] for reviewer SZg4), and different LLM architecture and pruning levels (see response [RQ1], [RQ2] for reviewer DVNq). All modifications to the paper are highlighted in blue. We hope these revisions address the reviewers' concerns and contribute to a more favorable assessment.

---

### Meta-Review · Area_Chair_aoTf · 2024-12-19

**Metareview:**

Reviews on this paper were split. Many agreed that the CoNNect regularizer was a conceptually interesting and novel method. The main drawback was that reviewers were not convinced by the provided experiments of the method's empirical performance, particularly of its performance as compared to state-of-the-art methods like Wanda and SparseGPT.

During the rebuttal period, the authors did a nice job of providing many additional experimental results to help address these concerns. However, reviewers were still not fully convinced. The authors pointed out several reasons why, despite some lower accuracy metrics, their method may still be applicable (support of structured pruning and better inference time speedups, ability to integrate with state-of-the-art methods, etc.)

Given these responses, I think this paper has potential and would encourage the authors to edit the paper to include many of the new experimental results and full comparisons with state-of-the-art methods and to focus on tackling one of the key problems they mention. E.g., if the real gain of the method is on inference time speed-ups as opposed to raw sparsity, this should be highlighted and rigorously justified. If the key aspect is that the method is very general, then it should be combined with state-of-the-art methods to improve their performance.

**Additional Comments On Reviewer Discussion:**

See discussion in main meta review.

---

### Decision · Program_Chairs · 2025-01-22

Reject